# ALS/FTD-associated FUS activates GSK-3β to disrupt the VAPB–PTPIP51 interaction and ER–mitochondria associations

Radu Stoica[1,¶], Sébastien Paillusson[1,¶], Patricia Gomez-Suaga[1], Jacqueline C Mitchell[1], Dawn HW Lau[1], Emma H Gray[1,†], Rosa M Sancho[1,‡], Gema Vizcay-Barrena[2], Kurt J De Vos[1,§], Christopher E Shaw[1], Diane P Hanger[1], Wendy Noble[1] & Christopher CJ Miller[1,*]

## Abstract

Defective FUS metabolism is strongly associated with amyotrophic lateral sclerosis and frontotemporal dementia (ALS/FTD), but the mechanisms linking FUS to disease are not properly understood. However, many of the functions disrupted in ALS/FTD are regulated by signalling between the endoplasmic reticulum (ER) and mitochondria. This signalling is facilitated by close physical associations between the two organelles that are mediated by binding of the integral ER protein VAPB to the outer mitochondrial membrane protein PTPIP51, which act as molecular scaffolds to tether the two organelles. Here, we show that FUS disrupts the VAPB–PTPIP51 interaction and ER–mitochondria associations. These disruptions are accompanied by perturbation of $Ca^{2+}$ uptake by mitochondria following its release from ER stores, which is a physiological read-out of ER–mitochondria contacts. We also demonstrate that mitochondrial ATP production is impaired in FUS-expressing cells; mitochondrial ATP production is linked to $Ca^{2+}$ levels. Finally, we demonstrate that the FUS-induced reductions to ER–mitochondria associations and are linked to activation of glycogen synthase kinase-3β (GSK-3β), a kinase already strongly associated with ALS/FTD.

**Keywords** amyotrophic lateral sclerosis; frontotemporal dementia; glycogen synthase kinase-3β; protein tyrosine phosphatase interacting protein 51; vesicle-associated membrane protein-associated protein B
**Subject Categories** Membrane & Intracellular Transport; Molecular Biology of Disease

## Introduction

Mitochondria and the endoplasmic reticulum (ER) form tight structural associations such that ~5–20% of the mitochondrial surface is closely apposed (10–30 nm distances) to specialized regions of the ER termed mitochondria-associated ER membranes (MAM) [1–3]. These associations regulate a number of fundamental physiological functions including $Ca^{2+}$ and phospholipid exchange between the two organelles, energy metabolism and ATP production, mitochondrial biogenesis and trafficking, apoptosis, ER stress responses and autophagy [1–3]. The mechanisms by which MAM become associated with mitochondria are not properly understood but electron microscopy (EM) studies reveal the presence of structures that appear to tether the two organelles [4]. Recently, the integral ER protein, vesicle-associated membrane protein-associated protein B (VAPB), was shown to bind to the outer mitochondrial membrane protein, protein tyrosine phosphatase interacting protein 51 (PTPIP51), to form at least some of these tethers [5,6]. Thus, modulating VAPB and/or PTPIP51 expression induces appropriate changes in ER–mitochondria contacts and also $Ca^{2+}$ exchange between the two organelles, which is a physiological read-out of ER–mitochondria contacts [5,6].

Many of the functions regulated by ER–mitochondria contacts are perturbed in ALS and associated FTD. ALS is the most common form of motor neuron disease, and FTD is the second most common form of presenile dementia after Alzheimer's disease [7]. These diseases are now known to be clinically, genetically and pathologically linked [7,8]. Thus, damage to mitochondria, ATP production, $Ca^{2+}$ homoeostasis, lipid metabolism, axonal transport, autophagy and the ER including activation of the unfolded protein response (UPR) are all features of ALS/FTD, and ER–mitochondria communications impact upon all of these processes [1–3,9,10]. Indeed, recent studies have demonstrated that some ALS/FTD-associated insults

1   Department of Basic and Clinical Neuroscience, Institute of Psychiatry, Psychology and Neuroscience, Kings College London, London, UK
2   Centre for Ultrastructural Imaging, King's College London, London, UK
    *Corresponding author. Tel: +44 207 8480393; Fax: +44 207 7080017; E-mail: chris.miller@kcl.ac.uk
    ¶These authors contributed equally to this work
    †Present address: Multiple Sclerosis Society, London, UK
    ‡Present address: Alzheimer's Research UK, Cambridge, UK
    §Present address: Sheffield Institute for Translational Neuroscience, University of Sheffield, Sheffield, UK

disrupt ER–mitochondria contacts and associated functions. These include the expression of Tar DNA-binding protein-43 (TDP-43) and loss of Sigma-1 receptor (Sigma-R1) function [6,11]. Mutations in both *TDP-43* and *Sigma-R1* cause some familial forms of ALS/FTD and accumulations of TDP-43 are a major pathology of ALS/FTD [12–16].

Defects in fused in sarcoma (FUS) metabolism are strongly implicated in both ALS and FTD. FUS accumulations are a pathological feature in a significant number of ALS/FTD cases, mutations in *FUS* cause some familial forms of ALS and FTD, and overexpression of wild-type and ALS/FTD-mutant FUS induces aggressive disease in transgenic rodents [7,17–27].

FUS is a predominantly nuclear protein where it functions in DNA repair, transcription and splicing but a proportion is also normally present in the cytoplasm [26,27]. However, the mechanisms by which FUS induces disease are not clear and both gain and loss of function hypotheses have been proposed [26,27]. Here, we show that the expression of both wild-type and ALS-mutant FUS disrupt ER–mitochondria associations and that this is accompanied by reductions in binding of VAPB to PTPIP51. We also demonstrate that FUS perturbs cellular $Ca^{2+}$ homoeostasis and mitochondrial ATP production. Damage to mitochondria is strongly linked to ALS [28–34]. Finally, we show that FUS activates glycogen synthase kinase-3β (GSK-3β) and that GSK-3β is a regulator of ER–mitochondria associations. GSK-3β is already strongly implicated in ALS/FTD [6,35–37]. Thus, our findings reveal a new pathogenic mechanism for FUS involving activation of GSK-3β and disruption to ER–mitochondria associations.

# Results

### Wild-type and mutant FUS disrupt ER–mitochondria associations and the VAPB–PTPIP51 interaction

To determine the effects of FUS on ER–mitochondria associations, we quantified ER–mitochondria contacts in NSC34 motor neuron cells transfected with either enhanced green fluorescent protein (EGFP) control vector, EGFP-FUS or familial ALS mutants EGFP-FUSR521C or EGFP-FUSR518K. Several previous studies have utilized EGFP-tagged FUS [24,38] but to confirm that the EGFP-FUS was functional, we monitored the expression of endogenous FUS 72 h post-transfection. FUS displays an autoregulatory function such that overexpression by transfection reduces endogenous gene expression [38]. At this time point, we detected a marked decrease in endogenous FUS expression in both wild-type and mutant EGFP-FUS-transfected cells (Fig EV1). These findings are in agreement with previous studies, which also showed that the EGFP tag does not affect the autoregulatory function of FUS [38].

The EGFP tags were then used to isolate transfected cells using a cell sorter and ER–mitochondria associations quantified by determining the proportion of the mitochondrial surface that was closely apposed (< 30 nm) to ER following analyses by EM. This approach has been used previously [4,6,39]. Transfection of FUS did not lead to changes in the expression of the ER–mitochondria tethering proteins VAPB or PTPIP51, or mitofusin-2, which has been proposed as a further ER–mitochondria tether [40] (Fig 1A). Moreover, we detected no change in the numbers of mitochondria or ER

profiles in the presence of either wild-type or mutant FUS. However, compared to control cells, the expression of wild-type and mutant FUS all led to significant reductions in ER–mitochondria associations (Fig 1B).

We also enquired whether loss of FUS influenced ER–mitochondria associations. To do so, we treated NSC34 cells with control or FUS siRNAs and again monitored ER–mitochondria associations by EM. siRNA knockdown of FUS did not alter the expression of VAPB, PTPIP51 or mitofusin-2, and loss of FUS had no effect on ER–mitochondria associations (Fig 1C and D).

To complement the EM studies, we monitored the effects of FUS on ER–mitochondria associations using super resolution structured illumination microscopy (SIM) [41]. NSC34 cells were again transfected with EGFP control vector, EGFP-FUS, EGFP-FUSR521C or EGFP-FUSR518K and then immunostained for protein disulphide isomerase (PDI) and translocase of the outer mitochondrial membrane protein-20 (TOM20) to label ER and mitochondria, respectively. Co-localization analyses of PDI and TOM20 revealed that wild-type and mutant FUS again disrupted ER–mitochondria associations (Fig 2A).

Finally, we used *in situ* proximity ligation assays [42] to monitor the effects of FUS on ER–mitochondria associations. Here, conventionally fixed cells and tissues are probed with primary antibodies followed by secondary antibodies coupled to specific oligonucleotides. If the distances between antigens are small enough, these oligonucleotides facilitate hybridization and ligation of connector oligonucleotides to form a circular DNA molecule, which then serves as a template for rolling circular amplification. Use of labelled nucleotides enables microscopic detection and quantification of hybridization signals. The distances detected by proximity ligation assays are similar to those detected by resonance energy transfer between fluorophores (i.e. ~10 nm) [42]. For these assays, we used antibodies to the ER protein VAPB and the outer mitochondrial membrane protein PTPIP51 since VAPB and PTPIP51 interact directly to tether ER with mitochondria [5,6]. Proximity ligation assays have already been used to quantify ER–mitochondria associations and the VAPB–PTPIP51 interaction [5,11,43].

To demonstrate the specificity of the proximity ligation assays, we first performed experiments involving omission of VAPB, PTPIP51 or both VAPB and PTPIP51 primary antibodies. In agreement with previous studies [5], omission of VAPB and/or PTPIP51 antibodies produced very few signals whereas inclusion of these primary antibodies generated robust signals (Fig EV2A). We then monitored how FUS expression affected the VAPB–PTPIP51 interaction. Compared to control EGFP, transfection of either wild-type or mutant FUS did not affect the size of the cells (% size: EGFP 100 ± 5.93, EGFP-FUS 96.86 ± 4.57, EGFP-FUSR521C 102 ± 5.3, EGFP-FUSR518K 112.9 ± 5.9; analysed by one-way ANOVA). However, compared to controls, reduced interactions between VAPB and PTPIP51 were detected in both wild-type and mutant FUS-transfected NSC34 cells (Fig 2B). Thus, in three different assays, wild-type and ALS/FTD-mutant FUS disrupt ER–mitochondria associations.

As a complement to these cellular studies, we investigated ER–mitochondria associations in spinal cord motor neurons of 10-week-old homozygous FUS transgenic mice and their non-transgenic littermates [19]. These transgenic mice express wild-type FUS under control of the commonly used mouse prion gene

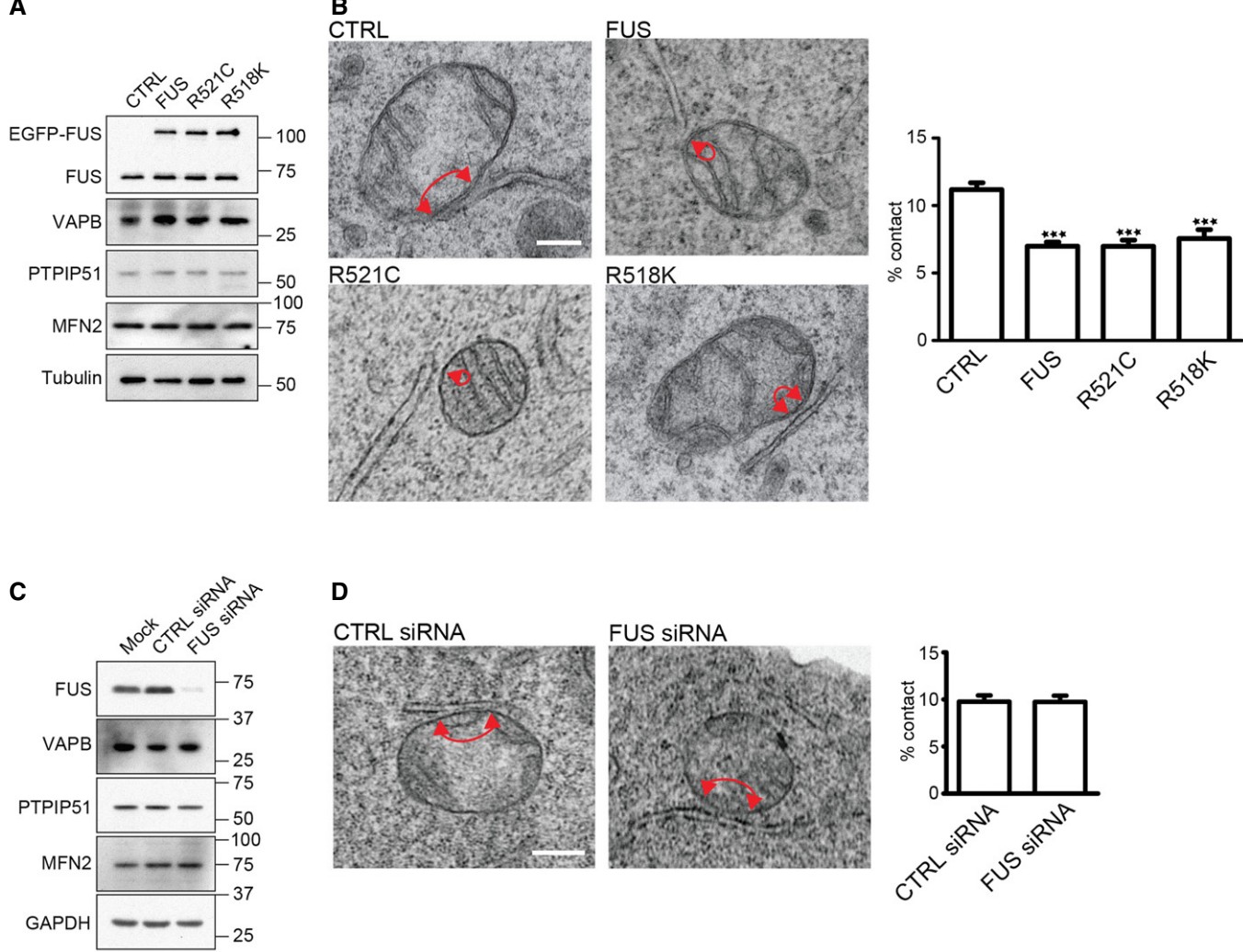

**Figure 1. Expression of wild-type and ALS/FTD-mutant FUS reduces ER–mitochondria associations in NSC34 cells.**

A    Expression of FUS does not alter expression of VAPB, PTPIP51 or mitofusin-2 (MFN2) in transfected NSC34 cells. Immunoblots of NSC34 cells transfected with EGFP as a control (CTRL), or wild-type or mutant EGFP-FUS. Transfected cells were purified via EGFP using a cell sorter and the samples probed on immunoblots as indicated. On the FUS immunoblot, samples were probed with FUS antibody to show endogenous and transfected proteins; tubulin is shown as a loading control.

B    Representative electron micrographs of ER–mitochondria associations in NSC34 cells transfected with control EGFP vector (CTRL), EGFP-FUS, EGFP-FUSR521C or EGFP-FUSR518K as indicated; arrowheads with loops show regions of association. Scale bar = 200 nm. Bar chart shows % of the mitochondrial surface closely apposed to ER in the different samples. Data were analysed by one-way analysis of variance (ANOVA) followed by Tukey's multiple comparison test. $N$ = 27–30 cells and 247–424 mitochondria, error bars are s.e.m.; ***$P$ < 0.001.

C, D    siRNA loss of FUS does not affect ER–mitochondria associations or alter expression of VAPB, PTPIP51 or mitofusin-2 (MFN2) in NSC34 cells. (C) Immunoblots of cells either mock transfected or treated with control (CTRL) or FUS siRNAs; GAPDH is shown as a loading control. (D) Representative electron micrographs of ER–mitochondria associations in control (CTRL) and FUS siRNA-treated cells. Arrowheads with loops show regions of association. Scale bar = 200 nm. Data analysed by unpaired $t$-test. $N$ = 27–28 cells and 193–202 mitochondria, error bars are s.e.m.

regulatory elements, which drive expression to a variety of neural cell types including motor neurons [19]. The homozygous FUS transgenic mice develop aggressive features of ALS including hindlimb paralysis and death by 12 weeks of age [19]. Similar to the findings in NSC34 cells, no changes in the expression of VAPB, PTPIP51 or mitofusin-2 were detected in FUS transgenic mice (Fig 3A). However, EM analyses revealed that compared to non-transgenic control littermates, ER–mitochondria associations were significantly reduced in spinal cord motor neurons of the FUS transgenics (Fig 3B). We also utilized VAPB/PTPIP51

proximity ligation assays to quantify ER–mitochondria contacts in spinal cord motor neurons of the FUS transgenic mice. Again, we first tested the specificity of the assays by performing control experiments in which VAPB, PTPIP51 or both VAPB and PTPIP51 primary antibodies were omitted and these revealed that omission of primary antibodies produced very few signals but their inclusion generated robust signals (Fig EV2B). Compared to control non-transgenic littermates, we detected no differences in motor neuron size in the FUS transgenics (% size: control 100 ± 4.96, FUS 95.65 ± 5.32; analysed by unpaired $t$-test). However, the

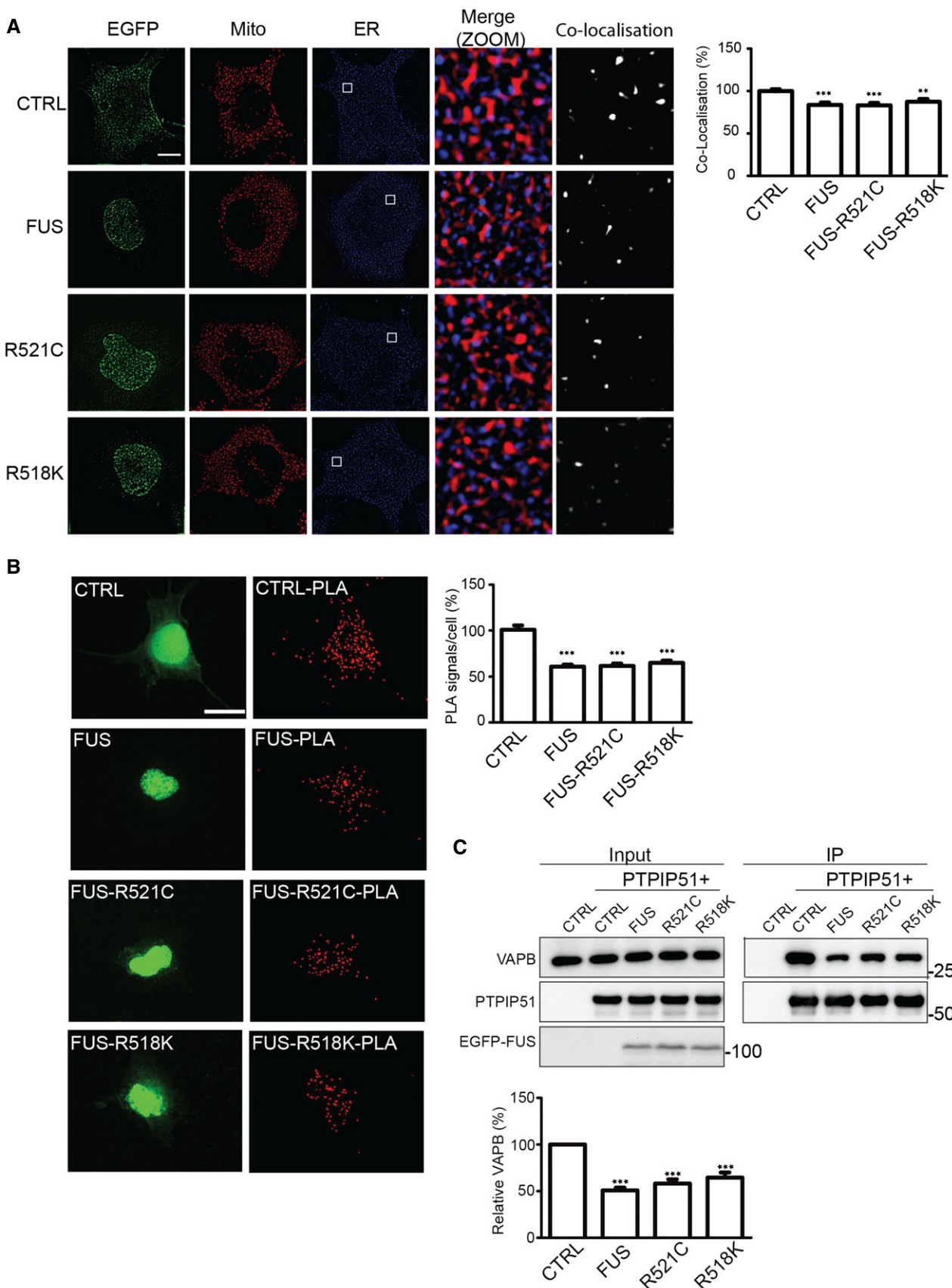

**Figure 2.**

**Figure 2.   Expression of wild-type and mutant FUS reduces ER–mitochondria associations and the VAPB–PTPIP51 interaction in NSC34 cells.**

A   FU-induced reductions in ER–mitochondria associations can be detected using SIM. NSC34 cells were transfected with either EGFP control vector, EGFP-FUS, EGFP-FUSR521C or EGFP-FUSR518K and immunostained for TOM20 and PDI to label mitochondria (Mito) and ER, respectively; FUS was detected via their EGFP tags. Merge (ZOOM) shows zoomed images of boxed regions, and co-localization shows co-localized pixels. Scale bar = 2 μm. Bar chart shows ER–mitochondria co-localization (Manders coefficient) normalized to control in the different samples. Data were analysed by one-way ANOVA with Tukey's *post hoc* test. A total of 10–14 cells were analysed per condition from three independent experiments; error bars are s.e.m., $**P < 0.01$ and $***P < 0.001$.

B   ER–mitochondria associations and the VAPB–PTPIP51 interaction are disrupted by wild-type and ALS/FTD-mutant FUS. NSC34 cells were transfected with EGFP control vector (CTRL), EGFP-FUS, EGFP-FUSR521C or EGFP-FUSR518K and proximity ligation assays performed using VAPB and PTPIP51 antibodies. FUS were detected via their EGFP tags. Scale bar = 10 μm. Bar chart shows relative number of proximity ligation assay signals/cell. Data were analysed by one-way ANOVA and Tukey's *post hoc* test; $n = 47–53$ cells from four experiments. Error bars are s.e.m.; $***P < 0.001$.

C   Overexpression of FUS reduces the binding of VAPB to PTPIP51 in transfected cells. Cells were transfected as indicated with either control empty vector (CTRL), HA-PTPIP51 + CTRL, or HA-PTPIP51 + either EGFP-FUS, EGFP-FUSR521C or EGFP-FUSR518K. PTPIP51 was immunoprecipitated using the HA tag and the amounts of endogenous bound VAPB detected by immunoblotting. Both inputs and immunoprecipitations (IP) are shown and no immunoprecipitating VAPB signals were obtained in the absence of HA-PTPIP51. Bar chart shows relative levels of VAPB bound to PTPIP51 in the immunoprecipitations following quantification of signals from immunoblots. VAPB signals were normalized to immunoprecipitated PTPIP51-HA signals. Data were analysed by one-way ANOVA and Tukey's *post hoc* test; $N = 4$. Error bars are s.e.m.; $***P < 0.001$.

VAPB/PTPIP51 proximity ligation assays revealed that in agreement with the EM analyses, ER–mitochondria associations were significantly reduced in spinal cord motor neurons of the FUS transgenics (Fig 3C).

Since we detected no FUS-induced changes in expression of VAPB or PTPIP51 in either NSC34 cells or transgenic mice (Figs 1A and C, and 3A), the proximity ligation assays not only confirm that FUS reduces ER–mitochondria associations but also show that this reduction involves breaking of the VAPB–PTPIP51 tethers. To test this further, we performed immunoprecipitation assays to monitor the affect of FUS on binding of VAPB to PTPIP51. To do so, we co-transfected cells with haemagglutinin (HA)-tagged PTPIP51 and either EGFP control vector, EGFP-FUS, EGFP-FUSR521C or EGFP-FUSR518K and monitored the amounts of VAPB bound to immunoprecipitated PTPIP51-HA by immunoblotting of the samples. Consistent with the proximity ligation assays, both wild-type and mutant FUS decreased the amounts of endogenous VAPB bound to immunoprecipitated PTPIP51-HA in these assays (Fig 2C). Thus, the FUS-induced reductions in ER–mitochondria associations involve decreased interactions between the tethering proteins VAPB and PTPIP51.

**Overexpression of FUS disturbs cellular Ca$^{2+}$ homoeostasis and mitochondrial ATP production**

A major function of ER–mitochondria associations is to regulate cellular Ca$^{2+}$ homoeostasis and in particular, to facilitate Ca$^{2+}$ exchange between ER and mitochondria following Ca$^{2+}$ release from ER stores [1–4,40]. Thus, disruption of the VAPB–PTPIP51 interaction perturbs cellular Ca$^{2+}$ homoeostasis [5,6]. Since expression of wild-type or mutant FUS reduced both ER–mitochondria and VAPB–PTPIP51 associations, we monitored the effect of overexpressing FUS on cytosolic and mitochondrial Ca$^{2+}$ levels after induction of inositol 1,4,5-trisphosphate (IP3) receptor-mediated Ca$^{2+}$ release from ER stores. For these experiments, we used HEK293 cells co-transfected with the M3 muscarinic acetylcholine receptor (M3R) and either control vector or FUS and triggered physiological IP3 receptor-mediated Ca$^{2+}$ release from ER stores by application of the M3R agonist oxotremorine-M. In line with previous studies on VAPB, PTPIP51 and ER–mitochondria associations, we used HEK293 cells for these experiments since they do not express endogenous M3R and so provide a useful

model for monitoring cytosolic and mitochondrial Ca$^{2+}$ levels after its release from ER specifically in transfected cells [5,6]. Transfection of both wild-type and mutant FUS all induced significant increases in cytosolic and decreases in mitochondrial Ca$^{2+}$ levels (Fig 4A and B). Such findings are consistent with the observed FUS-induced decreases in ER–mitochondria and VAPB–PTPIP51 interactions.

Damage to mitochondrial ATP production is a prominent feature of ALS [9,33,34,44]. Ca$^{2+}$ is required by mitochondria for generating ATP via the tricarboxylic acid cycle [45], and so, the reduced mitochondrial Ca$^{2+}$ levels seen in FUS-overexpressing cells predict that FUS impairs mitochondrial ATP production. We therefore monitored the effect of FUS-overexpression on mitochondrial ATP production using a FRET reporter system that permits ATP quantification in single living transfected cell [46]. Cellular ATP is generated by a combination of oxidative phosphorylation and glycolysis, and so, we assayed ATP production in cells treated with potassium cyanide (KCN), which inhibits cytochrome C oxidase to block oxidative phosphorylation. Monitoring ATP levels in the individually transfected cells prior to and after KCN treatment thus permits calculation of the levels of mitochondrial ATP production [46]. These studies revealed that compared to control transfected cells, both wild-type and mutant FUS reduced mitochondrial ATP production (Fig 4C).

**FUS activates GSK-3β**

To gain insight into the mechanism by which FUS might influence ER–mitochondria and VAPB–PTPIP51 associations, we first tested whether FUS bound to either VAPB or PTPIP51. Although FUS is a predominantly nuclear protein, a proportion is present in the cytoplasm where it might conceivably bind VAPB and/or PTPIP51 to disrupt their interaction [26,27]. To do so, we performed immunoprecipitation assays from FUS+VAPB and FUS+PTPIP51 transfected cells. However, we detected no binding of wild-type or ALS-mutant FUS to either protein in these assays (Fig EV3A and B).

Phosphorylation is a common mechanism for controlling protein–protein interactions and another ALS/FTD-associated protein; TDP-43 has been shown to activate GSK-3β to regulate binding of VAPB to PTPIP51 [6,35]. GSK-3β activity is regulated by a number of signalling pathways but a major route involves

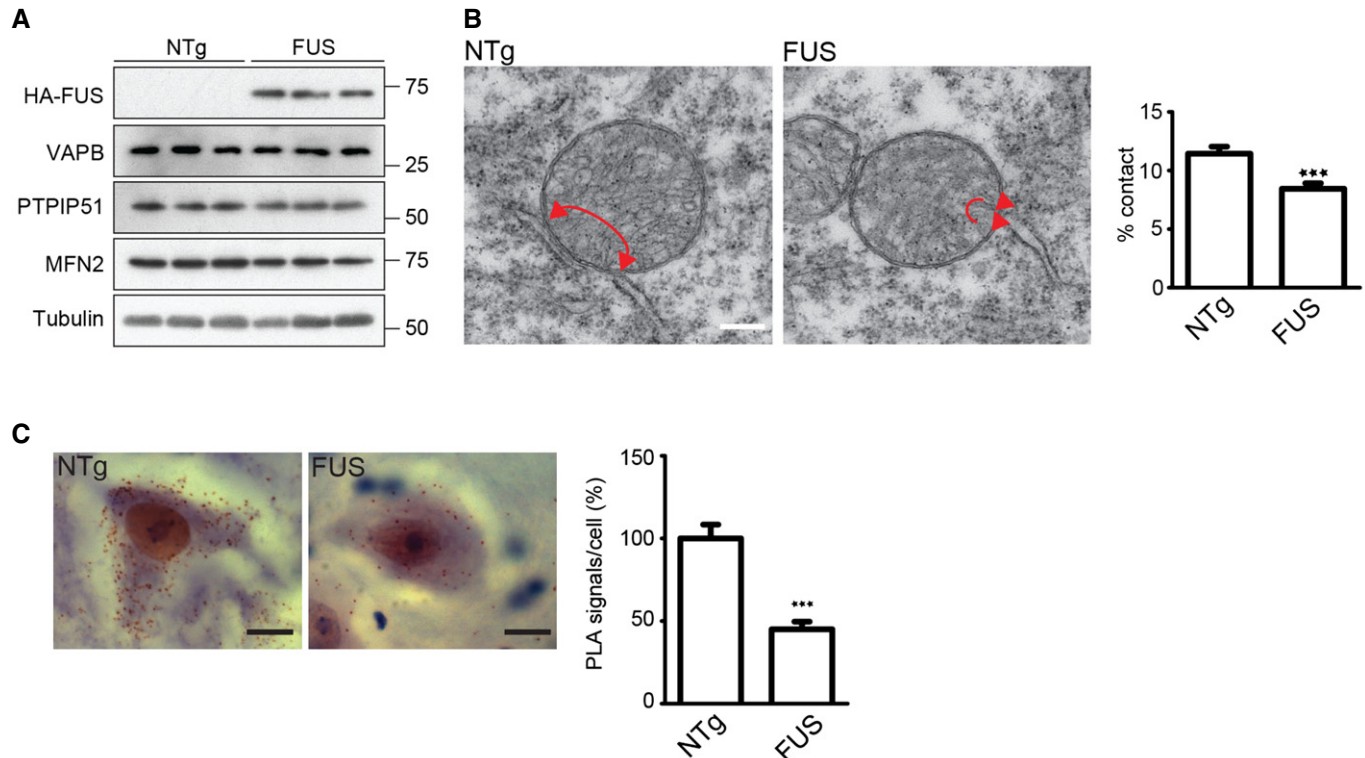

**Figure 3.  Overexpression of FUS reduces ER–mitochondria associations and the VAPB–PTPIP51 interaction in spinal cords of FUS transgenic mice.**

A   Overexpression of FUS does not affect expression of VAPB, PTPIP51 or mitofusin-2. Immunoblots of spinal cord proteins from three 10-week-old FUS transgenic mice and three age-matched littermates are shown; FUS was detected via its HA tag. Tubulin was used as a loading control.

B   Representative electron micrographs of ER–mitochondria associations in lumbar spinal cord motor neurons of FUS transgenic mice and their non-transgenic littermates; arrowheads with loops show regions of association. Scale bar = 200 nm. Bar chart shows % of mitochondrial surface closely apposed to ER in the two samples. Data were analysed by unpaired *t*-test. *N* = 67–88 cells and 438–749 mitochondria, error bars are s.e.m.; ***P* < 0.001.

C   ER–mitochondria associations and the VAPB–PTPIP51 interaction are disrupted in lumbar motor neurons in spinal cords of FUS transgenic mice. Representative images of proximity ligation signals in 11-week-old FUS and non-transgenic (NTg) littermate mice. Data were analysed by unpaired *t*-test; *N* = 40 cells from 3 FUS and 3 non-transgenic littermates (age 10 weeks). Error bars are s.e.m.; ***P* < 0.001.

inhibitory phosphorylation of serine 9 [47]. We therefore analysed GSK-3β serine 9 phosphorylation in control and FUS-transfected cells and in control and FUS transgenic mice by immunoblotting. Remarkably, we found that like TDP-43, overexpression of wild-type or ALS-mutant FUS activated GSK-3β as evidenced by reduced serine 9 phosphorylation (Fig 5A and B).

As detailed above, FUS is a predominantly nuclear protein but a proportion is present in the cytoplasm. The sequences governing FUS nuclear import reside at its extreme C-terminus [48,49]. To gain insight into whether the FUS-induced increases in GSK-3β activity and reduced binding of VAPB to PTPIP51 were linked to the levels of cytosolic FUS, we investigated how these features were affected by a mutant FUS in which the C-terminal nuclear localization signal was deleted (FUSΔC). Deletion of this nuclear localization signal has been shown to induce an almost complete redistribution of FUS to the cytosol [48,49]. Compared to wild-type FUS, cells transfected with FUSΔC displayed increased GSK-3β activity as evidenced by reduced serine 9 phosphorylation (Fig 5C). Moreover, this FUSΔC-induced increase in GSK-3β activity was accompanied by a significant decrease in binding of VAPB to PTPIP51 (Fig 5D). These findings support the notion that the

effects of FUS on the VAPB–PTPIP51 interaction may be linked to cytosolic FUS.

**Inhibition of GSK-3β promotes ER–mitochondria associations and the VAPB–PTPIP51 interaction**

GSK-3β has been shown to regulate binding of VAPB to PTPIP51 in biochemical assays; inhibition of GSK-3β increases the VAPB–PTPIP51 interaction [6]. To determine whether GSK-3β also regulates ER–mitochondria associations, we treated NSC34 cells with two structurally different GSK-3β inhibitors, AR-A014418 and CT99021, and quantified ER–mitochondria associations by EM. AR-A014418 and CT99021 are ATP competitive inhibitors so do not affect GSK-3β serine 9 phosphorylation. Both inhibitors significantly increased ER–mitochondria associations (Fig 6A). We also quantified the effects of the GSK-3β inhibitors on ER–mitochondria associations using VAPB–PTPIP51 proximity ligation assays. Neither inhibitor altered the size of the cells (% size: vehicle 100 ± 9.26, AR-A014418 91.16 ± 16.1, CT99021 115.4 ± 20.13; analysed by one-way ANOVA). However, both inhibitors again induced a significant increase in ER–mitochondria associations (Fig 6B). The

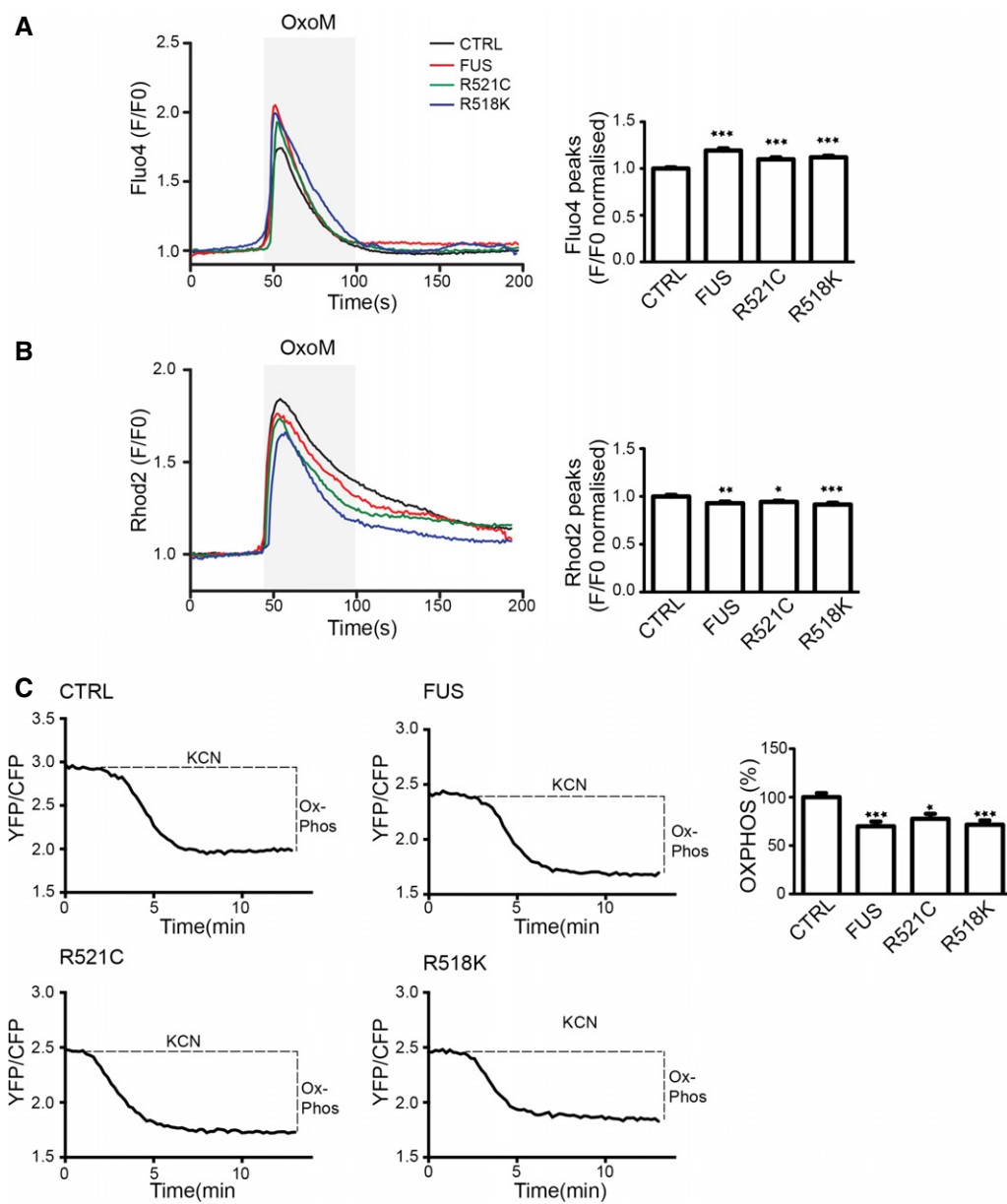

**Figure 4. Expression of FUS disrupts cellular Ca²⁺ homoeostasis and mitochondrial ATP production.**

A, B   FUS disrupts Ca²⁺ homoeostasis. HEK293 cells were transfected with M3R and either control vector (CTRL), FUS, FUSR521C or FUSR518K as indicated. Release of ER Ca²⁺ was induced by treatment of cells with OxoM. Panel (A) shows cytosolic Ca²⁺ levels with representative Fluo4 fluorescence traces on the left and normalized peak values on the right. Fluo4 fluorescence shows a transient increase in cytosolic Ca²⁺ levels upon OxoM treatment but compared to control, wild-type and mutant FUS all increase peak cytosolic Ca²⁺ levels. Panel (B) shows mitochondrial Ca²⁺ levels with representative Rhod2 fluorescence traces on the left and normalized peak values on the right. Data were analysed by one-way ANOVA and Tukey's *post hoc* test. (A) N = 49–52 cells from three experiments; (B) N = 50–52 cells from five experiments, error bars are s.e.m.; *P < 0.05, **P < 0.01, ***P < 0.001.

C   FUS reduces mitochondrial ATP production. ATP levels were measured in NSC34 cells transfected with the ATP indicator AT1.03 and either control vector (CTRL), HA-FUS, HA-FUSR521C or HA-FUSR518K. Cells were imaged in time-lapse prior to and after KCN treatment to inhibit oxidative phosphorylation. Representative traces of YFP/CFP ratios are shown for the different samples; initial YFP/CFP ratios prior to KCN treatment and those after KCN treatment are indicated. The fall in YFP/CFP ratios correlates with ATP produced by oxidative phosphorylation. Bar chart shows relative ATP levels produced by oxidative phosphorylation (OXPHOS) in the different samples. Data were analysed by one-way ANOVA and Tukey's *post hoc* test. N = 29–54 cells from five experiments, error bars are s.e.m.; *P < 0.05, ***P < 0.001.

VAPB–PTPIP51 proximity ligations assays also complement earlier biochemical assays involving immunoprecipitation assays, which likewise demonstrated that GSK-3β inhibitors increase binding of VAPB to PTPIP51 [6]. Thus and in agreement with previous studies, GSK-3β is a regulator of both ER–mitochondria associations and the VAPB–PTPIP51 interaction [6].

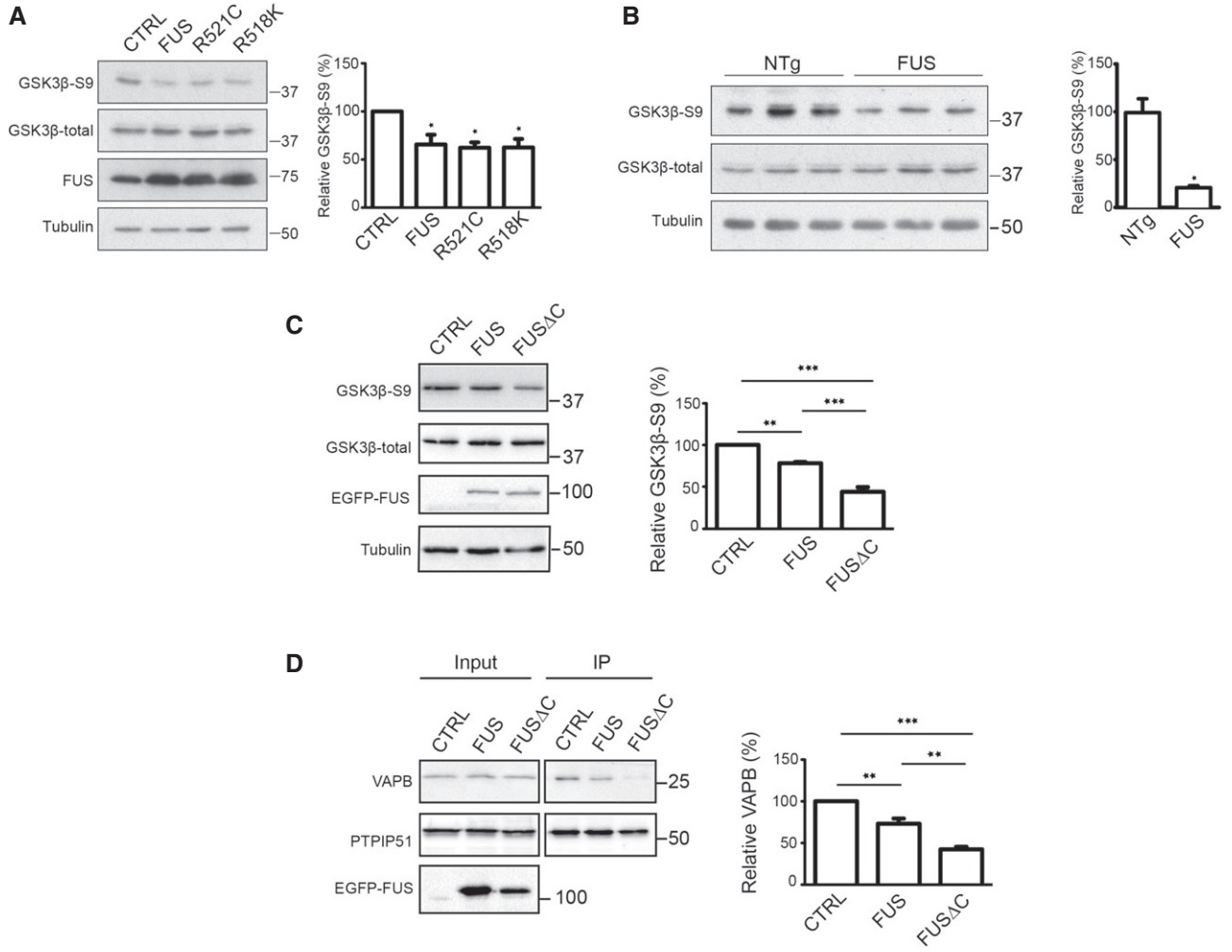

**Figure 5. FUS activates GSK-3β in transfected cells and transgenic mice.**

A Cells were transfected with either control vector (CTRL), HA-FUS, HA-FUSR521C or HA FUSR518K and the samples probed on immunoblots for GSK-3β phosphorylated on serine 9 (GSK-3β-S9), total GSK-3β, FUS (using FUS antibody) and tubulin as a loading control. Phosphorylation of GSK-3β serine 9 is the principal mechanism for regulating its activity; serine 9 phosphorylation inhibits GSK-3β activity. Bar chart shows relative levels of GSK-3β serine 9 phosphorylation following quantification of signals from immunoblots and normalization to total GSK-3β signals. Data were analysed by one-way ANOVA and Tukey's *post hoc* test. *N* = 4, error bars are s.e.m.; *\*P* < 0.05.

B Immunoblots of total and serine 9 phosphorylated GSK-3β in spinal cords from three 10-week-old FUS transgenic mice and their non-transgenic littermates. Samples were probed for tubulin as a loading control. Bar chart shows relative levels of GSK-3β serine 9 phosphorylation following quantification of signals from immunoblots and normalization to total GSK-3β signals. Data were analysed by unpaired *t*-test, error bars are s.e.m.; *\*P* < 0.05.

C Cytosolic FUS activates GSK-3β more potently than wild-type FUS. Cells were transfected with EGFP control, EGFP-FUS or EGFP-FUS lacking its C-terminal nuclear localization signal (FUSΔC). Samples were probed on immunoblots for total GSK-3β, GSK-3β phosphorylated on serine 9, FUS (using EGFP antibody) and tubulin as a loading control. Bar chart shows relative levels of GSK-3β serine 9 phosphorylation following quantification of signals from immunoblots and normalization to total GSK-3β signals. Data were analysed by one-way ANOVA and Tukey's *post hoc* test. *N* = 3, error bars are s.e.m.; *\*\*P* < 0.01, *\*\*\*P* < 0.001.

D Cytosolic FUS reduces the VAPB–PTPIP51 interaction more potently than wild-type FUS. Cells were transfected with PTPIP51-HA, and either EGFP control, EGFP-FUS or EGFP-FUSΔC. PTPIP51 was immunoprecipitated using the HA tag and bound endogenous VAPB detected by immunoblotting. Both inputs and immuno-precipitations (IP) are shown. FUS was detected using EGFP antibody. Bar chart shows relative levels of VAPB bound to PTPIP51 in the immunoprecipitations following quantification of signals from immunoblots. VAPB signals were normalized to immunoprecipitated PTPIP51-HA signals. Data were analysed by one-way ANOVA and Tukey's *post hoc* test; *N* = 3. Error bars are s.e.m.; *\*\*P* < 0.01, *\*\*\*P* < 0.001.

## Inhibition of GSK-3β corrects FUS-induced defects in ER–mitochondria associations and mitochondrial Ca²⁺ levels

The effects of the GSK-3β inhibitors on ER–mitochondria associations and the VAPB–PTPIP51 interaction prompted us to enquire whether inhibition of GSK-3β might correct defective ER–mitochondria associations induced by FUS. We therefore quantified ER–mitochondria associations by EM in FUS-expressing NSC34 cells treated with either vehicle or the GSK-3β inhibitor AR-A014418. Treatment with AR-A014418 abrogated the effect of FUS on ER–mitochondria associations (Fig 7). We also enquired whether AR-A014418 might correct the FUS-induced damage to

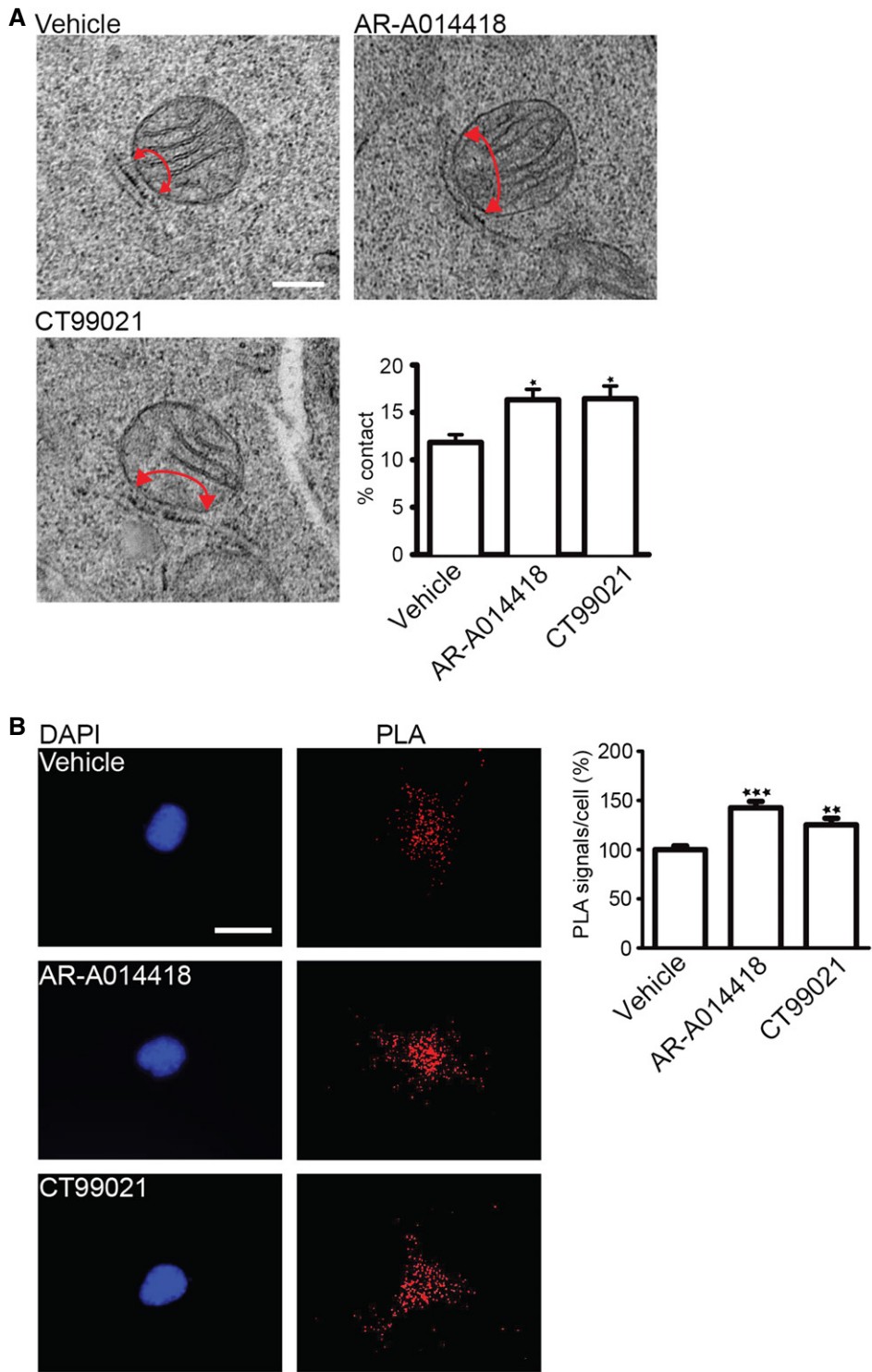

**Figure 6.** **Inhibition of GSK-3β increases ER–mitochondria associations and the VAPB–PTPIP51 interaction.**

A    Representative electron micrographs of ER–mitochondria associations in NSC34 cells treated with either vehicle or GSK-3β inhibitors AR-A014418 (1 μM) or CT99021 (100 nM) for 16 h. Arrowheads with loops show regions of association; scale bar = 200 nm. Bar charts shows % of the mitochondrial surface closely apposed to ER in the different samples. Data were analysed by one-way ANOVA followed by Tukey's multiple comparison test. *N* = 32 cells and 181–210 mitochondria. Error bars are s.e.m.; *$P$ < 0.05.

B    VAPB-PTPIP51 proximity ligation assays of NSC34 cells treated with either vehicle, 1 μM AR-A014418 or 100 nM CT99021 for 16 h. Cells were also stained for nuclei with DAPI. Bar chart shows relative number of proximity ligation assay signals/cell. Data were analysed by one-way ANOVA and Tukey's *post hoc* test; *n* = 103–173 cells from five experiments. Error bars are s.e.m.; **$P$ < 0.01, ***$P$ < 0.001.

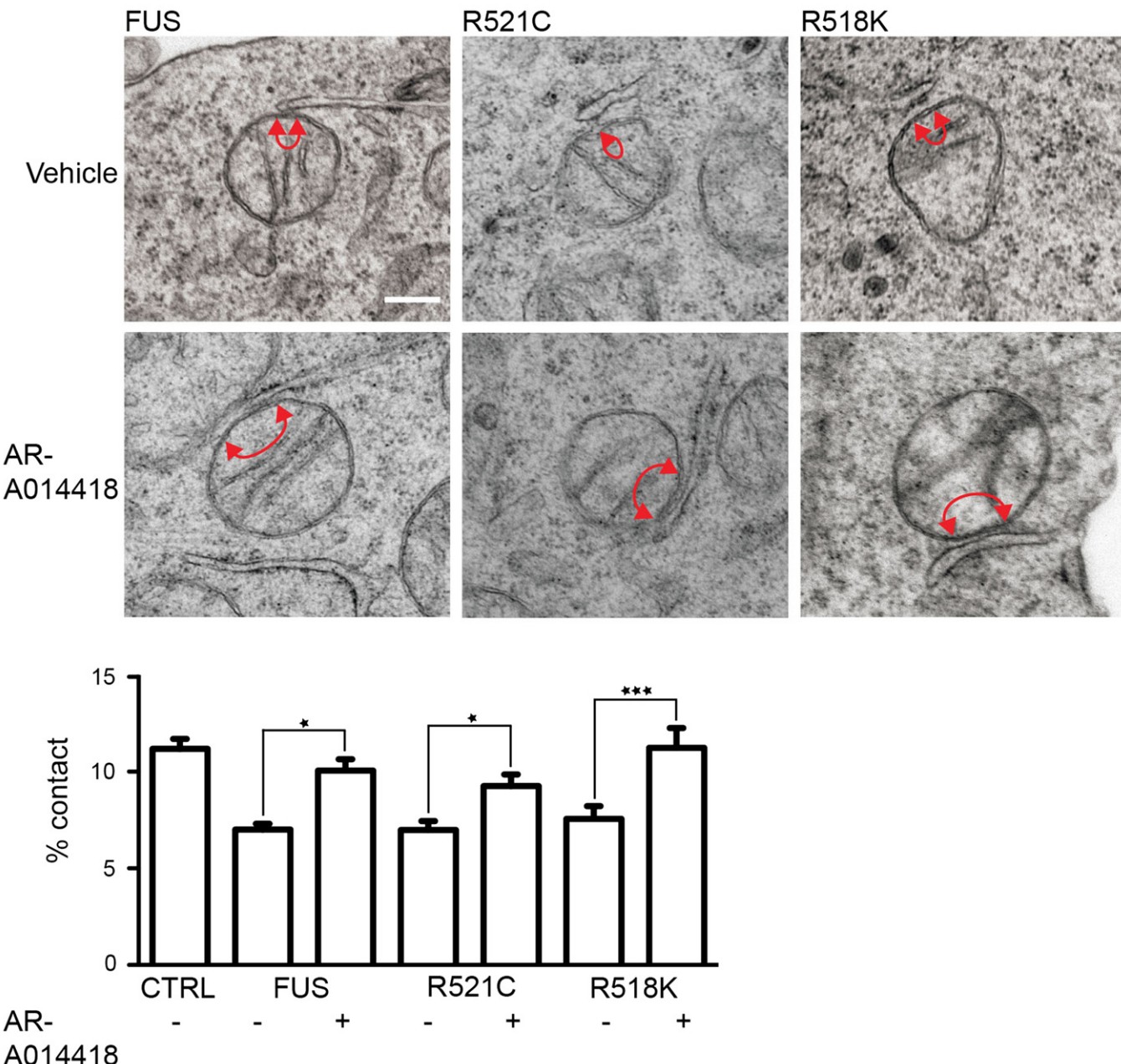

**Figure 7.  Inhibition of GSK-3β rescues defective ER–mitochondria associations induced by FUS.**
Representative electron micrographs of ER–mitochondria associations in NSC34 cells transfected with EGFP-FUS, EGFP-FUSR521C or EGFP-FUSR518K and treated with vehicle or 1 µM AR-A014418 for 16 h. Arrowheads with loops show regions of association; scale bar = 200 nm. Bar charts shows % of the mitochondrial surface closely apposed to ER in the different samples. Data were analysed by one-way ANOVA followed by Tukey's multiple comparison test. $N$ = 26–27 cells and 201–236 mitochondria. Error bars are s.e.m.; *$P$ < 0.05, ***$P$ < 0.001.

mitochondrial $Ca^{2+}$ uptake following its release from ER stores. To do so, we again used HEK293 cells co-transfected with M3R and either control vector or FUS and triggered physiological IP3 receptor-mediated $Ca^{2+}$ release from ER stores by application of oxotremorine-M. In line with the earlier experiments (see Fig 4B), wild-type and ALS/FTD-mutant FUS all induced a significant decrease in mitochondrial $Ca^{2+}$ levels. However, these decreases were abrogated by pre-treatment with AR-A014418 (Fig 8). Thus,

inhibition of GSK-3β rescues FUS-induced defects to ER–mitochondria associations and mitochondrial $Ca^{2+}$ levels.

## Discussion

Although FUS is a predominantly nuclear protein, a proportion is normally present in the cytoplasm where it has been linked to

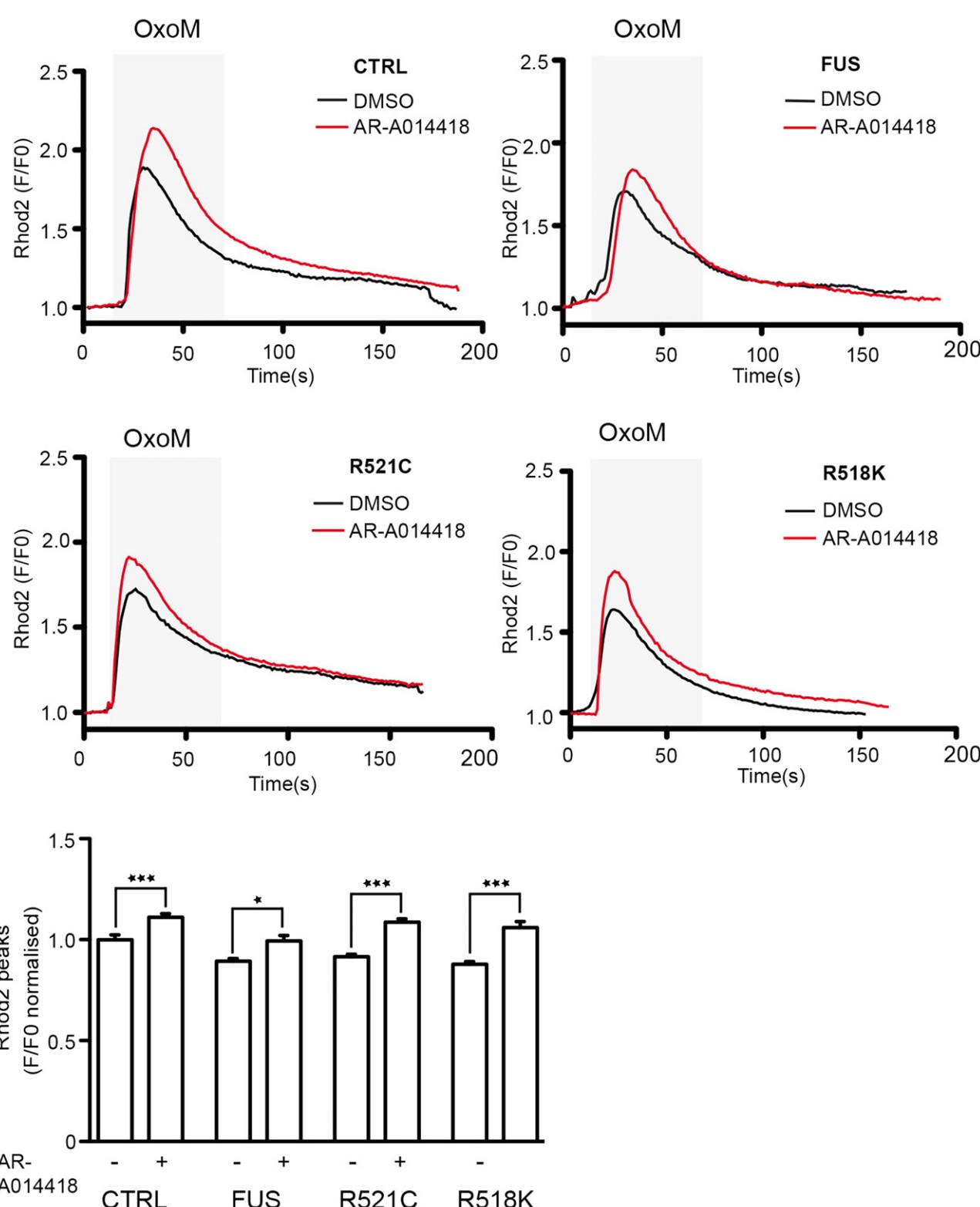

**Figure 8.  Inhibition of GSK-3β rescues FUS-induced defects in mitochondrial Ca²⁺ uptake following its release from ER stores.**
HEK293 cells were transfected with M3R and either control vector (CTRL), FUS, FUSR521C or FUSR518K and then treated with vehicle (DMSO) or 1 µM AR-A014418 for 16 h as indicated. Release of ER Ca²⁺ was induced by treatment of cells with OxoM. Representative Rhod2 fluorescence traces showing mitochondrial Ca²⁺ are shown along with bar chart displaying normalized peak values. Data were analysed by one-way ANOVA and Tukey's *post hoc* test. N = 30–100 cells from four experiments. Error bars are s.e.m.; *P < 0.05, ***P < 0.001.

mRNA translation in ribonucleoprotein complexes, to mRNA transport, mitochondrial function and in neurons, to the regulation of spine morphology [26,27,50–52]. However, FUS pathogenicity includes disturbances to mitochondria and to the ER [18,50,53,54]; such phenotypes suggest that FUS may damage the ER–mitochondria axis. To begin to test this possibility, we monitored the effect of FUS overexpression on ER–mitochondria associations. We show here that wild-type and familial ALS-mutant FUS decrease ER–mitochondria associations in both NSC34 motor neuron cells and transgenic mice.

Our findings that wild-type as well as mutant FUS decrease ER–mitochondria associations are consistent with the phenotypes observed in FUS transgenic mice where both wild-type and mutant FUS induce disease [18–23]. Indeed, in one study that compared directly transgenic mice expressing wild-type or ALS-mutant FUS-R521G at levels similar to endogenous mouse FUS, the mice expressing wild-type FUS displayed lower survival rates than the mutants [23]. The reasons behind this effect are not properly understood but there is evidence that mutations in the 3′ untranslated region (UTR) of *FUS* cause both familial and sporadic ALS by increasing wild-type FUS protein expression [55]. Recently, the effects of the 3′ UTR mutations have been shown to involve disruption to a FUS–microRNA regulatory circuit [56]. Thus, both clinical genetic evidence and experimental evidence involving multiple transgenic mouse studies demonstrate that overexpression of wild-type FUS induces disease. Interestingly, a similar feature may exist for TDP-43 in ALS/FTD since the expression of the wild-type protein likewise induces aggressive disease in mice and a 3′-UTR *TDP-43* variant linked to ALS also increases expression [57,58]. Like FUS, both wild-type and mutant TDP-43 disrupt ER–mitochondria associations to similar extents [6]. Indeed, the FUS and TDP-43 results in ALS/FTD complement findings in familial Alzheimer's and Parkinson's diseases where mutation and overexpression of wild-type amyloid precursor protein and α-synuclein involving increased gene dosage cause disease [59–63].

ER–mitochondria associations control $Ca^{2+}$ exchange between the two organelles, and this can impact upon the ability of mitochondria to generate ATP [1,45]. This is because the ER represents a primary source of mitochondrial $Ca^{2+}$ and several mitochondrial dehydrogenases in the TCA cycle are $Ca^{2+}$ regulated [45]. Our findings that FUS decreases mitochondrial $Ca^{2+}$ levels following IP3 receptor-mediated $Ca^{2+}$ release from ER stores and that this translates into reduced ATP levels is thus consistent with the loosening of ER–mitochondria contacts seen in FUS-expressing cells.

Damage to mitochondria is a major feature of ALS and a number of mutant proteins linked to familial ALS perturb mitochondrial function and/or biogenesis [9]. These include mutant Cu/Zn superoxide dismutase-1 (SOD1), TDP-43, optineurin, TANK-binding kinase-1 and valosin-containing protein (VCP) [28–33]. More recently, FUS too has been linked to mitochondrial damage [18,50,64,65]. A primary function of mitochondria is the generation of ATP, and reduced ATP levels are increasingly linked to the neurodegenerative process in ALS/FTD. For example, mutant SOD1, VCP and Sigma-R1 mutations all lower mitochondrial ATP production [33,44,66]. Indeed, elegant recent studies have demonstrated that relatively small reductions in ATP levels (similar to those we detect in the FUS-expressing cells) can induce many salient features of ALS [34]. Thus, the reductions in

ER–mitochondria interactions and associated disruptions to mitochondrial $Ca^{2+}$ levels and ATP production are likely to contribute to FUS pathogenesis.

VAPB and PTPIP51 represent tethering molecules, which interact to mediate ER–mitochondria associations [5,6], and we showed that the FUS-induced reductions to ER–mitochondria contacts involved reduced binding of VAPB to PTPIP51. Such findings support the notion that FUS targets the VAPB–PTPIP51 complex for damage. FUS is involved in transcription and mRNA metabolism, but we detected no changes in the levels of either VAPB or PTPIP51 in cells overexpressing wild-type or mutant FUS, nor in FUS siRNA knockdown cells. Thus, the FUS-induced changes in binding of VAPB to PTPIP51 are not the result of alterations to their expression.

As a further possibility, we tested whether FUS bound to either VAPB or PTPIP51; such binding might sterically inhibit the VAPB–PTPIP51 interaction. We detected no binding of wild-type or mutant FUS to either VAPB or PTPIP51 in immunoprecipitation assays, which suggest that FUS is not a major binding partner for VAPB or PTPIP51. However, we cannot eliminate the possibility that a proportion of FUS, beyond the sensitivity of our assays interacts with either VAPB or PTPIP51 and that this might contribute to changes in the VAPB–PTPIP51 interaction.

Finally, we monitored whether FUS might activate GSK-3β since GSK-3β has been shown previously to regulate binding of VAPB to PTPIP51; activation of GSK-3β decreases the VAPB–PTPIP51 interaction [6]. FUS activated GSK-3β in both transfected cells and in the FUS transgenic mice. Moreover, inhibition of GSK-3β increased ER–mitochondria associations and rescued the FUS-induced disruptions to ER–mitochondria associations and mitochondrial $Ca^{2+}$ levels. Thus, GSK-3β is a regulator of VAPB–PTPIP51 interactions and ER–mitochondria associations, and the effect of FUS on ER–mitochondria associations is linked to activation of GSK-3β. At this stage, the precise mechanism by which GSK-3β influences binding of VAPB to PTPIP51 is not clear; it may be via direct phosphorylation of VAPB and/or PTPIP51 to inhibit their binding, or by signalling to downstream effectors that somehow influence VAPB/PTPIP51 phosphorylation and/or binding. Future studies to identify and functionally characterize any VAPB and PTPIP51 phosphorylation sites would assist in discriminating between these possibilities.

Recently, two other familial ALS/FTD-linked proteins, TDP-43 and the Sigma-1 receptor have been shown to disrupt ER–mitochondria associations [6,11]. The Sigma-1 receptor is a chaperone that localizes to the regions of ER apposed to mitochondria (MAM) [3,11], but the mechanisms linking it to loosening of ER–mitochondria contacts and disease are not clear. However, the TDP-43-induced disruption to ER–mitochondria contacts is associated with reduced binding of VAPB to PTPIP51 that involves activation of GSK-3β [6]. Thus, three different ALS/FTD insults perturb ER–mitochondria associations and for FUS and TDP-43 at least, the mechanisms involve activation of GSK-3β.

One difficulty in deciphering ALS/FTD pathogenicity is that it involves damage to a large number of physiological processes. These include disruptions to $Ca^{2+}$ signalling, lipid and energy metabolism, mitochondria, the ER involving activation of the unfolded protein response, axonal transport and autophagy [9,10]. However, all of these cellular functions are regulated by ER–mitochondria associations [1–3,10]. Thus, damage to the ER–mitochondria axis

may provide an explanation for many of the seemingly disparate pathological features of ALS/FTD.

Recently, damage to ER–mitochondria associations have also been described in both Alzheimer's and Parkinson's diseases [43,67–73]. Perturbation of the ER–mitochondria axis may therefore be a general feature of all of the major human neurodegenerative diseases. As such, the results describe here and in these other reports highlight the ER–mitochondria axis as a new therapeutic target for neurodegenerative diseases.

# Materials and Methods

### Plasmids and siRNAs

Mammalian expression vectors for HA-PTPIP51, HA-FUS, HA-FUSR521C and myc-VAPB all in pCIneo, EGFP-FUS, EGFP-FUSΔC (FUSK510X), M3R, ATeam AT1.03 and control pCIneo vector expressing *E. coli* chloramphenicol acetyltransferase (CAT) have been described previously [5,6,46,48,74]. FUSR518K was created using a QuikChange XL mutagenesis kit (Stratagene). Mutagenic primers were 5′-CCAGGGGTGAGCACAAACAGGATCGCAGGGAG-3′ and 5′-CTCCCTGCGATCCTGTTTGTGCTCACCCCTGG-3′. Control and FUS siRNAs were from Dharmacon; FUS siRNA sequences were UCGACUGGUUUGAUGGUAA and GACUAUACCCAACAAGCAA and were applied together.

### Antibodies and inhibitors

Rat and rabbit antibodies to VAPB and PTPIP51 have been described previously and were generated by immunization with GST-VAPB(1–220) and GST-PTPIP51(36–470) [5]. Rabbit PTPIP51 antibody (FAM82A2) was from Atlas Antibodies. Rabbit anti-haemagglutinin (HA), mouse anti-α-tubulin (DM1A) and rabbit anti-mitofusin-2 were from Sigma. Mouse anti-myc (9B11) and rabbit anti-glyceraldehyde 3-phosphate dehydrogenase (GAPDH) were from Cell Signaling. Rabbit anti-FUS (NB100-565) was from Novus Biologicals. Rabbit anti-TOM20 was from Santa Cruz Biotechnology and mouse anti-PDI (RL77) was from Affinity Bioreagents. Antibodies to total and ser9 phosphorylated (inactive) GSK-3β were from BD Transduction Labs (mouse 610201) and Cell Signalling (rabbit 9336), respectively. Rabbit anti-GFP (Ab290) was from Abcam. GSK-3β inhibitors AR-A014418 and CT99021 were from Abcam and Cayman, respectively, and made up as 1 mM or 100 μM stocks in DMSO; KCN was from Sigma and made up as a 1 M stock in water.

### Cell culture and transfection

NSC34 cells were provided by Professor Dame Pamela Shaw (University of Sheffield, UK), and HEK293 cells were obtained from the ATCC. Cells were grown in Dulbecco's modified Eagle's medium containing 10% foetal bovine serum supplemented with 2 mM glutamine, 100 U/ml penicillin and 100 μg/ml streptomycin. NSC34 cells transfected with plasmids or siRNAs using Lipofectamine 2000 (Invitrogen) according to the manufacturer's instructions; HEK293 cells were transfected with Fugene 6 (Roche) according to the manufacturers' instructions. Except where stated, plasmid-transfected

cells were analysed 36 h post-transfection; siRNA-treated cells were analysed 3 days post-treatment. For fluorescence-activated flow cytometry cell sorting, NSC34 were trypsinized, resuspended in phosphate-buffered saline (PBS) and then sorted for EGFP expression using a BD FACSAria cell sorter, re-plated and cultured for a further 16 h prior to processing.

### Electron microscopy

For EM, NSC34 cells were fixed with 2.5% glutaraldehyde in 0.1 M sodium cacodylate buffer (pH 7.2) for 3 h and then harvested by scraping gently with a plastic scraper. The cells were pelleted by centrifugation at 800 *g* for 10 min, washed in buffer and post-fixed for 1 h in 1% osmium tetroxide in 0.1 M sodium cacodylate buffer. The cells were then stained for 1 h with 1% uranyl acetate in water before dehydration and embedding in TAAB resin. Mice were perfusion fixed in 2.5% glutaraldehyde/2% paraformaldehyde in PBS and spinal cords dissected and immersed in 2.5% glutaraldehyde/2% paraformaldehyde in 0.1 M sodium cacodylate buffer (pH 7.2) for 16 h. Samples were then post-fixed in 1% osmium tetroxide, dehydrated in alcohol and propylene oxide and finally embedded in Epon 81226. Sections were counterstained with lead citrate and uranyl acetate and samples viewed on a Tecnai 12 electron microscope at 6,800 or 9,300 magnification. Digital images were acquired and the circumference of each mitochondria and the proportions of the mitochondrial surface (circumference) closely associated (< 30 nm) with ER were calculated. Cells were randomly selected for analyses without prior knowledge of transfected plasmid, siRNA or mouse genotype. All clearly identified mitochondria in the samples were scored. Image analyses were performed using ImageJ and statistical analyses determined using GraphPad Prism.

### Super resolution structured illumination microscopy

NSC-34 cells plated on glass coverslips were fixed with 3% paraformaldehyde and 0.1% glutaraldehyde in phosphate-buffered saline (PBS) for 10 min at 20°C. After washing with PBS, samples were quenched by incubation with 50 mM NaBH$_4$ in PBS for 7 min, washed in PBS and then permeabilized for 30 min in PBS containing 0.2% Triton X-100 and 3% bovine serum albumin (BSA). Following blocking with 3% BSA in PBS, samples were incubated with primary antibodies diluted in blocking solution, washed with PBS and incubated with goat anti-rabbit or anti-mouse secondary Igs conjugated to AlexaFluor 546 or AlexaFluor 633 (1:1,000) (Invitrogen). Following final washings in PBS, the samples were mounted in Mowiol-DABCO mounting medium containing 10% (w/v) Mowiol 4-88 (Calbiochem), 25% (w/v) glycerol and 2.5% (w/v) DABCO (1,4-diazobicyclo[2.2.2]octane) in 100 mM Tris–HCl pH 8.5. Samples were analysed on a Nikon Eclipse Ti-E inverted microscope equipped with a Nikon N-SIM Super Resolution System. Confocal images were captured using a CFI Plan Apo IR SR 60X water immersion objective and then reconstructed using Nikon Imaging Software Elements AR with N-SIM module. ER–mitochondria interactions were quantified by Mander's coefficient using ImageJ software. Co-localized signals were displayed using the ImageJ 1.44p RG2B co-localization plug-in to determine co-localized pixels.

### Immunoprecipitations, SDS–PAGE and immunoblotting

Cells were harvested for SDS–PAGE and immunoblotting by scraping into SDS–PAGE sample buffer containing 2% SDS, 100 mM dithiothreitol, 10% glycerol, 0.1% bromophenol blue and protease inhibitors (Complete Roche) in 50 mM Tris–HCl pH 6.8 and heating to 100°C for 5 min. Transgenic mouse samples were homogenized in 50 mM Tris–citrate pH 7.4, 150 mM NaCl, 1% Triton X-100, 5 mM ethylene glycol tetraacetic acid, 5 mM EDTA and protease inhibitors (Complete, Roche) and then prepared for SDS–PAGE by addition of SDS–PAGE sample buffer and heating to 100°C for 5 min.

For immunoprecipitation assays, transfected cells were lysed in ice-cold immunoprecipitation buffer comprising 50 mM Tris–citrate pH 7.4, 150 mM NaCl, 1% Triton X-100, 5 mM ethylene glycol tetraacetic acid, 5 mM EDTA and protease inhibitors (Complete, Roche). The samples were then centrifuged at 13,000 *g* for 20 min, and the supernatants were transferred to fresh tubes and precleared by incubation with Protein-G-Sepharose beads (prepared as a 50% slurry in PBS containing 0.1% Triton X-100). Following centrifugation at 2,000 *g* for 30 s to settle the beads, supernatants were transferred to fresh tubes and protein concentrations were determined using a Bio-Rad protein assay kit. The protein concentrations were adjusted to 1 mg/ml, and 500 μg of protein was incubated with appropriate primary antibodies on a rotary shaker for 16 h at 4°C. Antibodies were then captured by addition of 30 μl Protein-G-sepharose beads (50% slurry in PBS/0.1% Triton X-100). Following washing in PBS/0.1% Triton X-100, bound proteins were prepared for SDS–PAGE by addition of 50 μl SDS–PAGE sample buffer and heating to 100°C.

Samples were separated on 10% (w/v) acrylamide gels and transferred to Protran nitrocellulose membranes (Schleicher & Schuell) using a Mini-PROTEAN 3 gel electrophoresis system and Transblot system (Bio-Rad). Following blocking in 5% (w/v) dried milk/0.1% w/v Tween-20 in TBS, the immunoblots were probed with primary antibodies diluted in blocking solution and after washing in blocking solution, incubated with horseradish peroxidase-conjugated goat anti-mouse, anti-rabbit or anti-rat Igs (GE Healthcare). Immunoblots were developed using an enhanced chemiluminescence system (GE Healthcare). Signals on immunoblots were quantified using ImageJ after scanning with an Epson Precision V700 Photo scanner essentially as described by us in previous studies [6]. To ensure the signals obtained were within the linear range, the mean background-corrected optical density (OD) of each signal was interpolated for an OD calibration curve created using a calibrated OD step tablet (Kodak). Only film exposures that gave OD signals within the linear range of the OD calibration curve were used for statistical analyses. Some signals were also quantified on a Bio-Rad ChemiDoc MP Imaging System.

### Proximity ligation assays

Proximity ligation assays to quantify VAPB–PTPIP51 interactions were performed essentially as described previously using Duolink reagents (Olink Bioscience) [5]. For cultured cells, cells were fixed in 4% paraformaldehyde in PBS and probed with rabbit anti-VAPB and rat anti-PTPIP51 antibodies. Signals were developed using the Duolink *In Situ* Orange kit; control samples were counterstained

with 4′,6-diamidino-2-phenylindole (DAPI) to show nuclei. For mouse spinal cords, paraformaldehyde perfusion fixed spinal cords were post-fixed in 4% paraformaldehyde containing 15% sucrose in PBS for 5 h, cryoprotected in 30% sucrose in PBS for 24 h and 30-μm sections were prepared using a cryostat. Samples were probed with rabbit anti-VAPB and rat anti-PTPIP51 antibodies. Signals were developed using the Duolink *In Situ* Brightfield kit and sections counterstained with Mayer's haematoxylin. NSC34 cells were imaged using a DM5000 fluorescence microscope (Leica Microsystems) equipped with a 40×/0.75NA HCX-PL-FLUOTAR lens; spinal cord sections were imaged using an Axioplan microscope (Zeiss) with Axiovision software. Cell sizes were determined using ImageJ and proximity ligation assay signals quantified using the particle analysis function of ImageJ. Statistical analyses were performed using GraphPad Prism.

### Ca²⁺ measurements

Cytosolic and mitochondrial $Ca^{2+}$ levels were measured following IP3 receptor-mediated $Ca^{2+}$ release from ER stores as previously described [5]. Briefly, HEK293 cells co-transfected with M3R and either empty vector or FUS was loaded with 2 μM Fluo4-AM or Rhod2-AM dye (Invitrogen) in external solution (145 mM NaCl, 2 mM KCl, 5 mM NaHCO₃, 1 mM MgCl₂, 2.5 mM CaCl₂, 10 mM glucose, 10 mM Na-HEPES pH 7.25) containing 0.02% Pluronic-F27 (Invitrogen) for 15 min at 37°C, followed by washing in external solution for 15 min. Fluo4 and Rhod2 fluorescences were timelapse recorded (1 s intervals) with MetaMorph (Molecular Dynamics) on an Axiovert S100 microscope (Zeiss) equipped with appropriate filter sets (Chroma Technology), a 40×/1.3NA Plan-Neofluar objective (Zeiss) and a Photometrics Cascade-II 512B EMCCD. The cells were kept under constant perfusion with external solution (0.5 ml/min). IP3 receptor-mediated $Ca^{2+}$ release from ER stores was triggered by application of 100 μM oxotremorine-M (Tocris) for 2 min. $Ca^{2+}$ levels were calculated as relative Fluo4 or Rhod2 fluorescence compared to baseline fluorescence at the start of the measurement.

For monitoring the effects of GSK-3β inhibition on mitochondrial $Ca^{2+}$ levels, HEK293 cells were co-transfected with M3R and either control vector, FUS, FUS-R521C or FUSR518K for 24 h and then treated with AR-A014418 for a further 16 h. $Ca^{2+}$ measurements were performed essentially as described above except using a Nikon Ti-E microscope using a CFI Plan Apo VC 20× objective and Nikon Andor Neo sCMOD high-resolution camera.

### ATP measurements

ATP levels in control and FUS-transfected cells were determined using a FRET-based plasmid reporter (Adenosine 5′-Triphosphate indicator based on Epsilon subunit for Analytical Measurements; ATeam reporter) [46]. To do so, NSC34 cells were co-transfected with AT1.03 cytosolic ATeam reporter and either control vector, FUS, FUSR521C or FUSR518K. Cells were imaged in Hanks' balanced salt solution (HBSS) without phenol red at 37°C by timelapse microscopy (12 s intervals) on a Zeiss Axiovert S100 microscope equipped with a 40×/1.3NA Plan-Neofluar objective and a Photometrics Cascade-II 512B EMCCD driven by MetaMorph (Molecular Dynamics). FRET filtersets (ECFP excitation filter ET

430/24×; ECFP emission filter 470/24 m; EYFP emission filter ET545/40 m) were from Chroma Technology. KCN 1 mM in HBSS was applied using a peristaltic pump (0.5 ml/min). YFP/CFP ratios prior to and after KCN treatment were measured as described and used to calculate relative ATP levels in the different samples, which were displayed as bar charts [46].

**Expanded View** for this article is available online.

## Acknowledgements

This work was supported by grants from the UK Medical Research Council, Alzheimer's Research UK, The Wellcome Trust, the Motor Neurone Disease Association, Parkinson's UK and the Rosetrees Trust. We also thank staff in the King's College Nikon Imaging Centre for microscopy assistance.

## Author contributions

CCJM, RS, SP, PG-S and KJDV designed the experiments. RS, SP, PG-S, DHWL and GV-B performed the experiments. JCM, EHG, RMS and CES provided reagents. CCJM, RS, SP, PG-S, DHWL, WN, DPH and KJDV wrote the manuscript.

## Conflict of interest

The authors declare that they have no conflict of interest.

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
