## [Review Process File · EMBO Reports]

Manuscript EMBO-2015-41726

ALS/FTD-associated FUS activates GSK-3 β to disrupt the VAPB-PTPIP51 interaction and ER-mitochondria associations

Radu Stoica, Sébastien Paillusson, Patricia Gomez-Suaga, Jacqueline Mitchell, Dawn Lau, Emma Gray, Rosa Sancho, Gema Vizcay-Barrena, Kurt De Vos, Christopher Shaw, Diane Hanger, Wendy Noble and Christopher Miller

Corresponding author: Christopher Miller, King's College London, Institute of Psychiatry, Psychology and Neuroscience

Review timeline:

Submission date:	10 November 2015
Editorial Decision:	14 January 2016
Revision received:	06 May 2016
Accepted:	13 June 2016

Editor: Barbara Pauly

Transaction Report:

1st Editorial Decision

14 January 2016

Thank you very much for the submission of your research manuscript to our editorial office. First of all, please accept my sincerest apologies for the unusual delay in getting back to you with a decision on your study. We have just now received the full set of reports from the referees that were asked to assess it. As the detailed reports are pasted below I will only repeat the main points here.

As you will see, all referees agree on the potential interest of the findings and in principle support its publication here. While referees 1 and 2 only raise rather minor concerns (which nevertheless would need to be addressed), referee 3 raises two related issues that I would like to point out specifically: First, this reviewer feels that a better characterization of the ER-mitochondrial contact sites in the cultured cells is necessary. Second and related to this, s/he feels that additional data from transgenic mice on the dysregulation of the ER/mitochondrial interaction would strengthen the in vitro findings. After further discussion of this point with the other referees, I would suggest that you perform at least initial analyses of these interactions in mice, and maybe compare the life spans of animals with and without GSK-beta inhibition. Please do let me know whether you think this is feasible.

Given the potential interest of your study and the constructive suggestions of the reviewers on how to improve it, I would like to give you the opportunity to revise your manuscript, with the understanding that the main concerns of the referees should be addressed. Acceptance of the manuscript will depend on a positive outcome of a second round of review and I should also remind

you that it is EMBO reports policy to allow a single round of major revision only and that, therefore, acceptance or rejection of the manuscript will depend on the completeness of your responses included in the next, final version of the manuscript.

Revised manuscripts should be submitted within roughly three months of a request for revision. If you feel that this period is insufficient for a successful submission of your revised manuscript I can potentially extend this period slightly.

Supplementary/additional data: If, in the course of the revision, you decide to display some data as supplementary information, please consider the following: All supplementary figures are now called Expanded View Figures and should be labelled and referenced as Figure EV1, Figure EV2 etc. in the main text of the manuscript. The files should be supplied as separate, high-resolution images. The legends for the EV figures should be incorporated in the main body of the text after the legends for the main figures. For more details, please refer to our guide to authors.

I look forward to seeing a revised form of your manuscript when it is ready. Should you decide to seek rapid publication somewhere else instead, I would appreciate a short note as well.

REFEREE REPORTS

Referee #1:

The major finding of this manuscript is that GSK-3 β regulates ER-mitochondria associations and that inhibition of GSK3 β rescues defective FUS-mediated ER-mitochondria associations.

In 2014, Stoica et al. published similar findings as above in ALS/FTD-associated TDP-43. It has been known that GSK-3 β is strongly implicated in ALS and inhibition of GSK-3 β has already been proposed as a therapeutic candidate in ALS (Yang YM et al., 2013).

This study targets ALS specialized community in the first plan, but it will certainly increase our understanding of ER-mitochondria interplay and FUS pathology in general.

The study is not of a high novelty, but authors made very good work of testing and connecting already known facts within the new study in FUS model of ALS. The claims are convincing and well discussed. The experimental data are of sufficient quality.

Inhibition of GSK-3 β corrects defective ER-mt associations induced by FUS, but it is not clear if this rescue strategy have effect on disturbed Ca²⁺ homeostasis in FUS. An experiment answering this question would be worth to perform.

Referee #2:

Investigation of ER-mito contacts in neurodegeneration models is reasonable, particularly for ALS where VAPB is one of the many candidate genes. The authors used EGFP-FUS transfected cells as well as FUS transgenic mice to detect less ER-mito contact and reduced VAPB-PTPIP51 interaction. In the transfected cells, small but significant Ca²⁺ flow from ER stores into mitochondria and reduced respiratory activity was noted. Finally, enhanced GSK-3 β activity was indirectly detected by reduced serine-9 phosphorylation in cells and mice. Pharmacological inhibition of GSK-3 β restored the ER-mito contacts in the EGFP-FUS transfected cells. The study is straightforward, and although the effects look small, they appear not only significant but also meaningful. Nevertheless, a few major problems deserve closer attention:

1) ALS mutant FUS causes no significantly different effects as wild-type. Moreover, FUS silencing does not cause a converse increase in ER-mito contacts. While unfortunate, that does not disqualify the relevance of the present work in my opinion. There could be technical explanations, such as much longer-term pathogenic effects in human patients, and FUS not being limiting for the control of ER-mito contacts, respectively. Nevertheless, a more critical discussion of relevance seems appropriate.

2) Figure 1a shows no auto-inhibition of endogenous FUS by the EGFP-FUS constructs. How can we know that the transfected FUS fusion proteins are authentically active?

3) The authors themselves acknowledge that the mechanism by which FUS influences the mitochondrial readouts as well as GSK-3 β activity remain entirely unknown. We cannot be even sure that the effects are directly due to FUS or some kind of secondary cellular response. The authors suggest vaguely cytosolic FUS effects. For this purpose, it may be revealing to use more strongly nuclear import impaired FUS mutants, such as P525L or deltaC.

Additional, Minor Comments:

4) The authors state in the second introductory paragraph that "damage to mitochondria, ATP production, Ca²⁺ homeostasis, lipid metabolism, axonal transport, autophagy and the ER including activation of the unfolded protein response (UPR) are all features of ALS/FTD" and build the rationale for the present study on this very broad concept. The last concluding paragraph then extends that all this applies to Alzheimer's and Parkinson's diseases as well. Such dramatizing statements are bordering triviality, as practically every noxious cellular event has been tied to all neurodegenerative mechanisms at some point. Also the sentence "Damage to mitochondrial ATP production is believed to be a major driver of disease in ALS" (last paragraph introduction) applies to almost any neurological condition, doesn't it? Instead of superficially lumping everything together a more pointed writing style would be preferable, at least for my taste.

5) Figure 7 is a (mistaken?) duplicate of Figure 6b and can be deleted.

6) Page numbering missing.

Referee #3:

In general, the paper is well written and easy to follow. The authors provide supporting references to back up most of their claims of what their data suggests. The main issue is in their reliance on using 2D EM as a measure of ER-mitochondrial interactions. It is difficult to know if the selected images capture actual physiology in the cell. Images of mitochondria next to a membrane are shown and the assumption is that this membrane is the ER, but it could be any single membrane organelle. 2D images may not capture the true nature of the interactions between ER and mitochondria. ER markers using immune-EM may help identify the ER membrane with greater certitude. It is certainly understandable why this approach was used, but are the differences biological sufficient to cause neurodegeneration? Though appealing, is there hard data that ALS or ALS/FTD is a disorder of mitochondria? Ref 27 is a computer simulation and not actual data from ALS or FTD fibroblast derived neuronal cells or robust animal models. The paper stated that "For instance, in ALS, motor neurons that innervate the fast-fatigable (FF) motor units are affected early, whereas those that innervate the slow (S) motor units are affected late (Kanning et al., 2010). This does not suggest a primary mitochondrial pathology, rather the opposite. The Discussion should be therefore more nuanced. Addition of mouse data would strengthen the paper. Do the FUS mice show this kind of dysregulation of the ER and mitochondrial interaction and do GSK-B inhibitors rescue the pathology

Other points:

- The figure panels are lower case, the legends are upper cases. I think there is a typo in the figure legend of figure 1F. I think this should read as FUS transgenic mice rather than TDP43 transgenic mice. As it is written it conflicts with the text in the results section and the figure itself.
- I wonder for figure 2C and 2D, if there should be some normalization to the size of the cell. Obviously, the smaller the cell, the less signal that will be observed. I don't think that it is taken into consideration.
- With regard to figures 5A and B, I don't think they can make the assumption that the proteins don't bind based on a negative IP result. Who knows if the conditions were appropriate to allow for

detection of an interaction? Given the reported low abundance of FUS in the cytoplasm, it would be an impressive feat if they did demonstrate an interaction. FUS is reported in the bead proteome so it surprising it did not come down

- In figure 5C, if they transfected untagged FUS, the band in the control lane when probing with the FUS antibody makes sense. A concern is that that in their methods and all other blots mentioned either HA- or EGFP- tagged FUS and then one would expect a band of higher molecular weight than the endogenous FUS in the control lane (see figure 1A). The figure legend doesn't mention any tag on FUS, so I suppose benefit of the doubt is order?

- Figure 6B and figure 7 are the same. I think figure 7 is included by mistake, it is not mentioned in the text.

1st Revision - authors' response

06 May 2016

Thank you for forwarding me the Referees helpful comments on our manuscript. We have positively addressed these comments and our revised manuscript includes much new data with 7 additional Figure panels (new Figure 2A, Figure 2C, Figure 5C, Figure 5D, Figure 6B, Figure 8, and EV1). Our original Figures 2A and 2B which display control experiments to demonstrate the specificity of the proximity ligations assay have been moved to EV2. Likewise our original Figures 5A and 5B which display negative results showing no binding of FUS to either VAPB or PTPIP51 in immunoprecipitation experiments have been moved to EV3. Below, we detail our responses to the Referees comments.

Referee 1

1) Inhibition of GSK-3 β corrects defective ER-mt associations induced by FUS, but it is not clear if this rescue strategy have effect on disturbed Ca²⁺ homeostasis in FUS. An experiment answering this question would be worth to perform.

Our initial submission demonstrated that FUS disrupts ER-mitochondria associations and also uptake of Ca²⁺ by mitochondria following its release from ER stores. We also demonstrated that the GSK-3 β inhibitor AR-A014418 corrects the FUS-induced defects in ER-mitochondria associations. In the revised manuscript, we now show that this correction to ER-mitochondria associations by AR-A014418 is accompanied by rescue of defective mitochondrial Ca²⁺ levels. This data is presented in a new figure (Figure 8) and described in Results page 9 paragraph 2.

Referee 2

1) ALS mutant FUS causes no significantly different effects as wild-type. Moreover, FUS silencing does not cause a converse increase in ER-mito contacts. While unfortunate, that does not disqualify the relevance of the present work in my opinion. There could be technical explanations, such as much longer-term pathogenic effects in human patients, and FUS not being limiting for the control of ER-mito contacts, respectively. Nevertheless, a more critical discussion of relevance seems appropriate.

In the Discussion we addressed this issue by pointing out that expression of wild-type and ALS/FTD mutant FUS *both* cause aggressive disease in transgenic mice, and that mutations in the 3' untranslated region (UTR) of FUS that cause increased expression of wild-type FUS protein are linked to familial and sporadic ALS. In the revised manuscript, we have re-worked this part of the Discussion to strengthen and clarify this point. We point out that in one study which compares transgenic mice expressing identical levels of wild-type and ALS/FTD mutant FUS(R521G), the wild-type mice actually display lower survival rates than the mutants (see Figure 1 in this publication) [1]. Moreover, the disease-associated 3' UTR mutations have now been more closely linked to FUS overexpression via micro-RNAs and we cite this work in the revised Discussion [2]. To, summarise, there is strong evidence that overexpression of wild-type and mutant FUS are equally toxic. In this context, it is therefore not surprising that we detect no differences in ER-mitochondria associations between the wild-type and the ALS/FTD mutants. We hope this reworked section will clarify this issue for Referee 2. Of course, mutations in APP and α -synuclein cause familial Alzheimer's and Parkinson's diseases respectively but later studies showed that gene duplication/triplication of these genes to increase expression of wild-type protein can also cause familial disease. It may be that a similar situation will be revealed for FUS in future studies but of course identifying these genetic causes is more difficult than identifying mutations. For comparison,

we draw attention to the APP and α -synuclein data. These reworked aspects are presented in Discussion page 10 paragraph 2.

2) *Figure 1a shows no auto-inhibition of endogenous FUS by the EGFP-FUS constructs. How can we know that the transfected FUS fusion proteins are authentically active?*

We have analysed FUS expression in the EGFP-FUS transfected cells at a later time point (72 hours) and here we detect reduced expression of endogenous FUS consistent with autoregulation of the endogenous gene. This finding is in line with previous studies that likewise demonstrated autoregulation with EGFP-tagged FUS constructs [3]. This new data is presented as Expanded View Figure EV1 and described in Results (Page 5 paragraph 1).

3) *The authors themselves acknowledge that the mechanism by which FUS influences the mitochondrial readouts as well as GSK-3 β activity remain entirely unknown. We cannot be even sure that the effects are directly due to FUS or some kind of secondary cellular response. The authors suggest vaguely cytosolic FUS effects. For this purpose, it may be revealing to use more strongly nuclear import impaired FUS mutants, such as P525L or deltaC.*

FUS nuclear localization is regulated by C-terminal sequences and deletion of these sequences dramatically increases cytosolic FUS levels. To gain insight into whether the effects of FUS overexpression on GSK-3 β activity are linked to cytosolic FUS levels, Referee 2 suggests utilising a mutant in which these C-terminal sequences are deleted (deltaC; FUS Δ C). In the revised manuscript, we include new data showing that compared to wild-type FUS, FUS Δ C decreases binding of VAPB to PTPIP51 and is more potent at activating GSK-3 β (Figures 5C and 5D and described in Results page 8 paragraph 4). These findings support the notion that the effects of FUS on ER-mitochondria associations are linked to increased cytosolic levels of the protein. This was a really good suggestion by Referee 2 as it provides insight into mechanisms; we thank him/her for this input.

Since part of these data involve immunoprecipitation assays to show the effect of FUS Δ C on binding of VAPB to PTPIP51, we felt it necessary to include similar immunoprecipitation data on the effects of wild-type and ALS/FTD mutant FUS on the VAPB-PTPIP51 interaction (the earlier version relied on PLAs only). Our revised version therefore includes a new Figure to include this data. This new Figure shows that both wild-type and mutant FUS disrupt binding of VAPB to PTPIP51 in immunoprecipitation assays which is in agreement with the PLAs. These new data are displayed in Figure 2C and described in Results page 7 paragraph 2.

Additional, Minor Comments:

4) *The authors state in the second introductory paragraph that "damage to mitochondria, ATP production, Ca²⁺ homeostasis, lipid metabolism, axonal transport, autophagy and the ER including activation of the unfolded protein response (UPR) are all features of ALS/FTD" and build the rationale for the present study on this very broad concept. The last concluding paragraph then extends that all this applies to Alzheimer's and Parkinson's diseases as well. Such dramatizing statements are bordering triviality, as practically every noxious cellular event has been tied to all neurodegenerative mechanisms at some point. Also the sentence "Damage to mitochondrial ATP production is believed to be a major driver of disease in ALS" (last paragraph introduction) applies to almost any neurological condition, doesn't it? Instead of superficially lumping everything together a more pointed writing style would be preferable, at least for my taste.*

Referee 2 is correct in stating that practically every noxious cellular event has been tied to all neurodegenerative diseases. However, damage to mitochondria, the ER (activation of the UPR), Ca²⁺ homeostasis, axonal transport and autophagy are particularly strongly linked to disease and especially to ALS/FTD (see for e.g. [4]). Remarkably, all of these features are regulated by ER-mitochondria associations. Inflammation is also a prominent disease feature and inflammatory responses have now also been linked to ER-mitochondria associations with the discovery that the inflammasome forms at ER-mitochondria contact sites [5]. Such findings have stimulated interest in investigating ER-mitochondria contacts in neurodegenerative diseases and this is why we focused the rationale for our study on this concept. Indeed, since we submitted our manuscript, two major reviews have highlighted further, the relationship between ER-mitochondria contacts and neurodegenerative diseases [6, 7]. So whilst we take on board Referee 2s point (we have altered the Discussion relating to mitochondrial damage and ATP production in ALS; page 10 paragraph 4 and page 11, we hope to persuade him/her that the styling we have taken is valid.

5) *Figure 7 is a (mistaken?) duplicate of Figure 6b and can be deleted.*

We apologise for this mistake; the duplicated Figure has been deleted.

6) *Page numbering missing.*

We have inserted page numbers in the revised manuscript.

Referee 3

The main issue is in their reliance on using 2D EM as a measure of ER-mitochondrial interactions. It is difficult to know if the selected images capture actual physiology in the cell. Images of mitochondria next to a membrane are shown and the assumption is that this membrane is the ER, but it could be any single membrane organelle. 2D images may not capture the true nature of the interactions between ER and mitochondria. ER markers using immune-EM may help identify the ER membrane with greater certitude.

We are puzzled by this comment since ER is of course routinely identified in EM studies based upon morphology without the need for immune-labelling. Indeed, most of the images we show of ER actually display regions containing ribosomes so are clearly identifiable as ER. Standard (non-immune) EM methods have also been used to analyse ER contacts with other organelles such as mitochondria and this includes recent studies in high impact journals. Without providing an exhaustive list, some recent publications that utilise standard EM (without immune EM) to analyse ER-organelle contacts include work in *Developmental Cell*, *Nature Medicine*, *PNAS*, *J Clinical Investigation* and our own work in *Nature Communications* [8-12].

Nevertheless, we do take on board Referee 3's point. We have tried using immune-EM with our and with commercial VAPB and PTPIP51 antibodies without success. In our hands, fixation conditions that preserve ultrastructure result in loss of immunostaining and relaxing of fixation conditions to preserve immunogenicity give poor ultrastructure that does not permit robust quantification of contacts involving VAPB-PTPIP51 interactions. One aspect which we did not highlight properly in our manuscript is that our use of proximity ligation assays (PLAs) with VAPB and PTPIP51 antibodies also provides quantitative data on ER-mitochondria associations (VAPB and PTPIP51 are ER and mitochondrial proteins respectively and so serve as markers for these organelles). PLAs have a resolution similar to that of Förster resonance energy transfer which is 5-10 nm [13] and so are suitable for quantification of ER-mitochondria contacts. Moreover, PLAs have already been used to quantify changes in ER-mitochondria associations and so there are precedents for their use in this context [14, 15]. In the revised manuscript, we make it clearer that our PLAs complement the EM work and provide an alternative assay to quantify ER-mitochondria contacts. These changes are in the Results page 5 paragraph 5 and page 6 paragraphs 1-3. This re-work of the text has led to some alterations in the structuring of the Results section. We also include new PLA data which demonstrates that the GSK-3 β inhibitors increase ER-mitochondria contacts (new Figure 6B). Finally, in the revised manuscript, we also utilise super-resolution structured illumination microscopy (SIM) to analyse the effects of FUS on ER-mitochondria contacts and these experiments corroborate the EM assays and PLAs; this data is shown in a new Figure 2A. Thus, the revised version of our manuscript includes new experimental data and utilises three different assays, EM, PLAs and super resolution SIM to quantify ER-mitochondria associations. All of these assays demonstrate that FUS reduces ER-mitochondria contacts. We trust that Reviewer 3 will accept the robustness of these data.

It is certainly understandable why this approach was used, but are the differences biological sufficient to cause neurodegeneration? Though appealing, is there hard data that ALS or ALS/FTD is a disorder of mitochondria? Ref 27 is a computer simulation and not actual data from ALS or FTD fibroblast derived neuronal cells or robust animal models. The paper stated that "For instance, in ALS, motor neurons that innervate the fast-fatigable (FF) motor units are affected early, whereas those that innervate the slow (S) motor units are affected late (Kanning et al., 2010). This does not suggest a primary mitochondrial pathology, rather the opposite. The Discussion should therefore be more nuanced.

Referee 3 is correct that the recent *Neuron* publication we cite to link ALS to mitochondrial dysfunction is a computational model [16]. We cited this publication because it demonstrates a fundamental concept in that relatively small decreases in mitochondrial ATP production can trigger disease. However, we accept that just citing this work does not make the case for mitochondrial dysfunction as a disease mechanism in ALS. There is in fact, a wealth of data linking ALS with mitochondrial dysfunction (if you screen PUBMED with "mitochondria" "amyotrophic lateral sclerosis" "review"; you will find there are currently 260 REVIEWS that cover this topic). We have

therefore strengthened this part of the Discussion linking ALS with damage to mitochondria and ATP production. We point out that a number of mutant genes linked to ALS including SOD1, TDP-43, optineurin, TBK1 and VCP damage mitochondria, and that several of these impair ATP production. We also cite recent studies linking FUS with damage to mitochondria. This reworked section of the Discussion is now on page 10 paragraph 4 and page 11. We trust Referee 3 will accept this argument.

Addition of mouse data would strengthen the paper. Do the FUS mice show this kind of dysregulation of the ER and mitochondrial interaction and do GSK-3 inhibitors rescue the pathology.

In her covering letter, the Editor Dr Pauly amplified this point stating “*I would suggest that you perform at least initial analyses of these interactions in mice, and maybe compare the life spans of animals with and without GSK-3 β inhibition. Please do let me know whether you think this is feasible.*”

Firstly, it seems to have been missed but our original submission did actually contain data to show that ER-mitochondria contacts are reduced in FUS transgenic mice that develop ALS. These data are shown in Figures 3A, B and C (originally Figures 1E, F and 2D). We also show that GSK-3 β is activated in the FUS transgenic mice (Figure 5B; originally Figure 5D).

With regard to comparing lifespans of FUS transgenic mice after treatment with GSK-3 β inhibitors, then there are scientific and ethical reasons why we cannot perform these experiments. We discussed these issues with Dr Pauly shortly after receiving the Referees comments which she understood and stated that EMBO Reports would not insist on the addition of these data. However, for transparency Dr Pauly asked that we present our views on this point in the formal response to the Referees comments. We have therefore included our earlier comments as an Appendix at the end of this communication. We have also deleted the part of our Discussion relating to GSK-3 β inhibitors as a therapy since this was clearly misleading.

Other points:

- *The figure panels are lower case, the legends are upper cases. I think there is a typo in the figure legend of figure 1F. I think this should read as FUS transgenic mice rather than TDP43 transgenic mice. As it is written it conflicts with the text in the results section and the figure itself.*

We apologise for these mistakes which have now been corrected.

- *I wonder for figure 2C and 2D, if there should be some normalization to the size of the cell. Obviously, the smaller the cell, the less signal that will be observed. I don't think that it is taken into consideration.*

In the revised version, these Figures are now Figures 2B and 3C. There is also a further figure involving proximity ligation assays (Figure 6B). There are no significant differences between the size of control and FUS expressing cells (Figure 2B and 3C) or between vehicle and inhibitor treated cells (Figure 6B) in these assays. We present these data in relevant parts of the Results (page 6 paragraph 2 and 3; page 9 paragraph 1). Thus, differences in proximity ligation signals between treatments are not the result of changes in cell size.

- *With regard to figures 5A and B, I don't think they can make the assumption that the proteins don't bind based on a negative IP result. Who knows if the conditions were appropriate to allow for detection of an interaction? Given the reported low abundance of FUS in the cytoplasm, it would be an impressive feat if they did demonstrate an interaction. FUS is reported in the bead proteome so it surprising it did not come down.*

These figures have now been moved to Figure EV3. This is a valid point by Referee 3 and we have therefore altered our discussion of these findings so that they are more circumspect (see Discussion page 11 paragraph 3).

- *In figure 5C, if they transfected untagged FUS, the band in the control lane when probing with the FUS antibody makes sense. A concern is that that in their methods and all other blots mentioned either HA- or EGFP- tagged FUS and then one would expect a band of higher molecular weight than the endogenous FUS in the control lane (see figure 1A). The figure legend doesn't mention any tag on FUS, so I suppose benefit of the doubt is order?*

Figure 5C is now Figure 5A. We have now clarified the legend to state that it involves transfection

of HA-tagged FUS and probing with FUS antibody. Unlike the EGFP tag utilized in Figure 1A, the HA-tag involves addition of only 9 amino acids (YPYDVPDYA) and this does not noticeably affect its migration on SDS-PAGE in our hands.

• *Figure 6B and figure 7 are the same. I think figure 7 is included by mistake, it is not mentioned in the text.*

We apologise for this mistake and have removed this Figure.

Finally, we wish to thank all three Referees for their time and input. Irrespective of any decision by EMBO Reports to publish our study, their comments have helped us improve our manuscript.

Appendix -Communication to Dr Pauly 22.1.2016.

Dear Dr Pauly,

Thank you for getting back to me with the Reviewers comments and your very helpful advice on our manuscript. There was a little bit of a mix-up in receiving your email with the reviews but Joel in the Editorial office kindly sorted it out. In your email, you ask us to get back to you quickly about a point relating to GSK-3 β inhibitors and transgenic mice. So please forgive me for the little delay in responding; we obtained your email a bit later than I think was intended.

The point you requested our response to was raised by Reviewer 3 and involves testing the therapeutic potential of GSK-3 β inhibitors in the transgenic FUS mice; your suggestion was that we “perform at least initial analyses of these interactions in mice, and maybe compare the life spans of animals with and without GSK-3 β inhibition. Please do let me know whether you think this is feasible”.

Firstly, I note that Reviewer 3 is not insistent on testing of GSK-3 β inhibitors in the transgenic FUS mice but merely says such “mouse data would strengthen the paper”; but irrespective of this, I have to tell you that we cannot perform this particular experiment and I detail the reasons below.

1. GSK-3 β was a target for dementia treatment by the pharmaceutical industry and hence the development of inhibitors such AR-A014418 and CT99021 that we utilise in our study. Despite major effort by pharma, most GSK-3 β inhibitors failed in preclinical studies with only two making it to clinical trials; one an analogue of AR-A014418 (AstraZeneca compound) and NP-12/tideglusib. Clinical studies on AR-A014418 have been discontinued and tideglusib has recently failed to show efficacy (see [17-20]). One problem with targeting GSK-3 β for neurological disease is that it functions in many physiological processes outside of the nervous system including insulin signaling, wnt signaling, autophagy, apoptosis and of course its original identified role in glucose/glycogen metabolism [21]. Indeed, many of the inhibitors display liver toxicity including an AstraZeneca AR-A014418 analogue [22]; this is in agreement with the phenotype of GSK-3 β knockout mice which die due to liver degeneration [23]. Thus there is strong evidence that despite their effects on ER-mitochondria associations, treatment of the mice with the GSK-3 β inhibitors will not prove beneficial and this is probably due to effects of GSK-3 β outside of the nervous system.

2. Studies of lithium in ALS lend further support to a lack of beneficial affect of targeting GSK-3 β . Lithium, an FDA approved drug is a potent although non-specific inhibitor of GSK-3 β [24]. In a high profile publication, the therapeutic effects of lithium were studied in ALS transgenic mice and also in a patient cohort; marked beneficial affects were reported especially in the mice [25]. However, subsequent studies have failed to replicate any of these findings; properly powered and blinded experiments in the mice showed no improvement with lithium and neither did three full clinical trials in ALS patients [26-30].

3. The FUS mice we are using display aggressive disease and death by 12 weeks [31]. Your suggestion of experiments to determine the efficacy of the inhibitors in extending lifespan of the mice means allowing them to live to end-point. UK government Home Office regulations which govern experimental usage of mice such as you propose (see <https://www.gov.uk/guidance/research-and-testing-using-animals>) classify suffering as Mild, Moderate and Severe and taking the mice to end-point will of course be classified as Severe. Any experiments involving Severe banding fall under special Home Office restrictions that requires further ethical review and the time taken to review such work is not quick. Moreover, I believe it unlikely that permission will be given to us (or any other UK laboratory) to conduct animal experimentation banded as Severe that involves GSK-

3 β inhibitors where the approach has already failed in either preclinical or full clinical trials. As mentioned above, all trials for GSK-3 β inhibitors have failed in the clinic and many display toxicity [17-20, 22].

4. I quite understand EMBO Reports rules for a 3 month restricted time period to perform additional experiments. However, even if we had UK Home Office permission to perform the trial you propose, this could not be conducted within 3 months. The mice we are utilizing are FUS homozygous and as they die by 12 weeks [31], we have to breed them up when needed. Also, the “initial analyses” you indicate is not really scientifically rigorous and such work is not in line with current views on testing of therapeutics in transgenic mice. ARRIVE guidelines which we ascribe to involve testing of compounds in both transgenics and non-transgenics with properly powered Ns, monitoring of toxicity (which is a major issue with the GSK-3 β inhibitors) and of course performing detailed pharmacokinetics and relating these to any phenotype; in our case this would involve amounts of drug crossing the blood brain barrier [32]. To properly conduct any testing of a therapeutic in transgenic mice is a major experiment on its own and in the past, performing low number and powered studies have produced data that has not always been replicated; the lithium study that relates directly to GSK-3 β and ALS that I describe above is one such example.

To summarise, we are using the GSK-3 β inhibitors as experimental tools to probe mechanisms and not as potential therapeutics. As we detail above, there are both scientific and ethical reasons for not pursuing the trial of the GSK-3 β inhibitors you mention. Towards the end of our Discussion, we do briefly discuss the therapeutic potential of GSK-3 β inhibitors for FTD/ALS and if you are happy to let us resubmit without testing efficacy in mice, this is one aspect of our manuscript that we certainly need to alter. I suspect that our current discussion on this may well have prompted the request for testing!

References

1. Sephton CF, Tang AA, Kulkarni A, West J, Brooks M, Stubblefield JJ, Liu Y, Zhang MQ, Green CB, Huber KM, *et al.* (2014) Activity-dependent FUS dysregulation disrupts synaptic homeostasis. *Proc Natl Acad Sci USA* **111**: E4769-4778
2. Dini Modigliani S, Morlando M, Errichelli L, Sabatelli M, Bozzoni I (2014) An ALS-associated mutation in the FUS 3'-UTR disrupts a microRNA-FUS regulatory circuitry. *Nat Commun* **5**: 4335
3. Zhou Y, Liu S, Liu G, Ozturk A, Hicks GG (2013) ALS-associated FUS mutations result in compromised FUS alternative splicing and autoregulation. *PLoS Genet* **9**: e1003895
4. Ferraiuolo L, Kirby J, Grierson AJ, Sendtner M, Shaw PJ (2011) Molecular pathways of motor neuron injury in amyotrophic lateral sclerosis. *Nat Rev Neurol* **7**: 616-630
5. Zhou R, Yazdi AS, Menu P, Tschopp J (2011) A role for mitochondria in NLRP3 inflammasome activation. *Nature* **469**: 221-225
6. Krols M, van Isterdael G, Asselbergh B, Kremer A, Lippens S, Timmerman V, Janssens S (2016) Mitochondria-associated membranes as hubs for neurodegeneration. *Acta neuropathologica*
7. Paillusson S, Stoica R, Gomez-Suaga P, Lau DH, Mueller S, Miller T, Miller CC (2016) There's something wrong with my MAM; the ER-Mitochondria axis and neurodegenerative diseases. *Trends Neurosci* **39**: 146-157
8. Arruda AP, Pers BM, Parlakgul G, Guney E, Inouye K, Hotamisligil GS (2014) Chronic enrichment of hepatic endoplasmic reticulum-mitochondria contact leads to mitochondrial dysfunction in obesity. *Nat Med* **20**: 1427-1435
9. Manford AG, Stefan CJ, Yuan HL, Macgurn JA, Emr SD (2012) ER-to-plasma membrane tethering proteins regulate cell signaling and ER morphology. *Dev Cell* **23**: 1129-1140
10. Stoica R, De Vos KJ, Paillusson S, Mueller S, Sancho RM, Lau KF, Vizcay-Barrena G, Lin WL, Xu YF, Lewis J, *et al.* (2014) ER-mitochondria associations are regulated by the VAPB-PTPIP51 interaction and are disrupted by ALS/FTD-associated TDP-43. *Nat Commun* **5**: 3996
11. Filadi R, Greotti E, Turacchio G, Luini A, Pozzan T, Pizzo P (2015) Mitofusin 2 ablation increases endoplasmic reticulum-mitochondria coupling. *Proc Natl Acad Sci USA* **112**: E2174-E2181
12. Sepulveda-Falla D, Barrera-Ocampo A, Hagel C, Korwitz A, Vinueza-Veloz MF, Zhou K, Schonewille M, Zhou H, Velazquez-Perez L, Rodriguez-Labrada R, *et al.* (2014) Familial Alzheimer's disease-associated presenilin-1 alters cerebellar activity and calcium homeostasis. *J Clin Invest* **124**: 1552-1567

13. Soderberg O, Gullberg M, Jarvius M, Ridderstrale K, Leuchowius KJ, Jarvius J, Wester K, Hydbring P, Bahram F, Larsson LG, *et al.* (2006) Direct observation of individual endogenous protein complexes in situ by proximity ligation. *Nat Methods* **3**: 995-1000
14. Bernard-Marissal N, Medard JJ, Azzedine H, Chrast R (2015) Dysfunction in endoplasmic reticulum-mitochondria crosstalk underlies SIGMAR1 loss of function mediated motor neuron degeneration. *Brain : a journal of neurology* **138**: 875-890
15. Hedskog L, Pinho CM, Filadi R, Ronnback A, Hertwig L, Wiehager B, Larssen P, Gellhaar S, Sandebring A, Westerlund M, *et al.* (2013) Modulation of the endoplasmic reticulum-mitochondria interface in Alzheimer's disease and related models. *Proc Natl Acad Sci USA* **110**: 7916-7921
16. Le Masson G, Przedborski S, Abbott LF (2014) A computational model of motor neuron degeneration. *Neuron* **83**: 975-988
17. Martinez A, Perez DI, Gil C (2013) Lessons learnt from glycogen synthase kinase 3 inhibitors development for Alzheimer's disease. *Curr Top Med Chem* **13**: 1808-1819
18. Kramer T, Schmidt B, Lo Monte F (2012) Small-molecule inhibitors of GSK-3: structural insights and their application to Alzheimer's disease models. *Int J Alzheimers Dis* **2012**: 381029
19. Wolfe MS (2012) The role of tau in neurodegenerative diseases and its potential as a therapeutic target. *Scientifica* **2012**: 796024
20. Lovestone S, Boada M, Dubois B, Hull M, Rinne JO, Huppertz HJ, Calero M, Andres MV, Gomez-Carrillo B, Leon T, *et al.* (2015) A phase II trial of tideglusib in Alzheimer's disease. *J Alzheimers Dis* **45**: 75-88
21. Kaidanovich-Beilin O, Woodgett JR (2011) GSK-3: Functional insights from cell biology and animal models. *Front Mol Neurosci* **4**: 40
22. Hall AP, Escott KJ, Sanganee H, Hickling KC (2015) Preclinical toxicity of AZD7969: Effects of GSK3beta inhibition in adult stem cells. *Toxicol Pathol* **43**: 384-399
23. Hoeflich KP, Luo J, Rubie EA, Tsao MS, Jin O, Woodgett JR (2000) Requirement for glycogen synthase kinase-3b in cell survival and NF-kappaB activation. *Nature* **406**: 86-90
24. Stambolic V, Ruel L, Woodgett JR (1996) Lithium inhibits glycogen synthase kinase-3 activity and mimics Wingless signalling in intact cells. *Curr Biol* **12**: 1664-1668
25. Fornai F, Longone P, Cafaro L, Kastsiuchenka O, Ferrucci M, Manca ML, Lazzeri G, Spalloni A, Bellio N, Lenzi P, *et al.* (2008) Lithium delays progression of amyotrophic lateral sclerosis. *Proc Natl Acad Sci USA* **105**: 2052-2057
26. Aggarwal SP, Zinman L, Simpson E, McKinley J, Jackson KE, Pinto H, Kaufman P, Conwit RA, Schoenfeld D, Shefner J, *et al.* (2010) Safety and efficacy of lithium in combination with riluzole for treatment of amyotrophic lateral sclerosis: a randomised, double-blind, placebo-controlled trial. *Lancet Neurol* **9**: 481-488
27. Miller RG, Moore DH, Forshew DA, Katz JS, Barohn RJ, Valan M, Bromberg MB, Goslin KL, Graves MC, McCluskey LF, *et al.* (2011) Phase II screening trial of lithium carbonate in amyotrophic lateral sclerosis: Examining a more efficient trial design. *Neurology* **77**: 973-979
28. Morrison KE, Dhariwal S, Hornabrook R, Savage L, Burn DJ, Khoo TK, Kelly J, Murphy CL, Al-Chalabi A, Dougherty A, *et al.* (2013) Lithium in patients with amyotrophic lateral sclerosis (LiCALS): a phase 3 multicentre, randomised, double-blind, placebo-controlled trial. *Lancet Neurol* **12**: 339-345
29. Gill A, Kidd J, Vieira F, Thompson K, Perrin S (2009) No benefit from chronic lithium dosing in a sibling-matched, gender balanced, investigator-blinded trial using a standard mouse model of familial ALS. *PloS one* **4**: e6489
30. Pizzasegola C, Caron I, Daleno C, Ronchi A, Minoia C, Carri MT, Bendotti C (2009) Treatment with lithium carbonate does not improve disease progression in two different strains of SOD1 mutant mice. *Amyotrophic lateral sclerosis : official publication of the World Federation of Neurology Research Group on Motor Neuron Diseases* **10**: 221-8
31. Mitchell JC, McGoldrick P, Vance C, Hortobagyi T, Sreedharan J, Rogelj B, Tudor EL, Smith BN, Klasen C, Miller CC, *et al.* (2012) Overexpression of human wild-type FUS causes progressive motor neuron degeneration in an age- and dose-dependent fashion. *Acta Neuropathol* **125**: 273-288
32. Kilkenny C, Browne WJ, Cuthill IC, Emerson M, Altman DG (2010) Improving bioscience research reporting: the ARRIVE guidelines for reporting animal research. *PLoS Biol* **8**: e1000412

First of all I would like to apologise for the unusual delay in getting back to you with a decision on your manuscript. We have tried very hard to get in touch with the missing referees, without success. Therefore, and to keep you from waiting any longer, I am now making a decision based on the one report we received on the revised version and which was positive. I am thus very pleased to accept your manuscript for publication in the next available issue of EMBO reports. Thank you for your contribution to our journal.

Corresponding Author Name: Chris Miller

Manuscript Number: EMBOR-2015-41726V2